# Differentially Private Boxplots

**Kelly Ramsay** [* 1]   **Jairo Diaz-Rodriguez** [* 1]

## Abstract

Despite the potential of differentially private data visualization to harmonize data analysis and privacy, research in this area remains underdeveloped. Boxplots are a widely popular visualization used for summarizing a dataset and for comparison of multiple datasets. Consequentially, we introduce a differentially private boxplot. We evaluate its effectiveness for displaying location, scale, skewness and tails of a given empirical distribution. In our theoretical exposition, we show that the location and scale of the boxplot are estimated with optimal sample complexity, and the skewness and tails are estimated consistently, which is not always the case for a boxplot naively constructed from a single existing differentially private quantile algorithm. As a byproduct of this exposition, we introduce several new results concerning private quantile estimation. In simulations, we show that this boxplot performs similarly to a non-private boxplot, and it outperforms the naive boxplot. Additionally, we conduct a real data analysis of Airbnb listings, which shows that comparable analysis can be achieved through differentially private boxplot visualization.

## 1. Introduction

It is now well established that differential privacy (Dwork et al., 2006) is a powerful framework for protecting the privacy of individuals' data. As a result, a plethora of differentially private data analysis tools have been developed over the last several decades (Dwork, 2008; Ji et al., 2014; Liu et al., 2023). However, one area that has been considerably underdeveloped is that of differentially private visualization. This is despite the fact that data visualization is a key tool in exploratory analysis, which is an essential component of data analysis. In fact, in a recent study of data analysts and practitioners, Garrido et al. (2023) found that "most analysts employed aggregations and visualizations to fulfill their use case," rather than machine learning models.

Differentially private versions of several popular visualizations have been considered in the literature. Nanayakkara et al. (2022) developed a plot to visualize privacy utility trade-offs. Zhou et al. (2022) developed `DPVisCreator`, a visualization system, which relies on publishing a synthetic dataset. Visualizations in other privacy models were considered by Dasgupta & Kosara (2011); Dasgupta et al. (2013; 2019), and recently reviewed by Bhattacharjee et al. (2020). There has been substantial study of the differentially private histogram (Hay et al., 2010; Li et al., 2010; Acs et al., 2012; Kellaris & Papadopoulos, 2013; Xu et al., 2013; Zhang et al., 2014; Budiu et al., 2022). Visualizations based on clustering (Hongde et al., 2014), for mobility data (He et al., 2016), heatmaps (Zhang et al., 2016) and scatterplots (Panavas et al., 2024) have also been considered.

Surprisingly, despite its widespread use, a differentially private boxplot has not been directly studied. Boxplots, invented by Tukey et al. (1977), are used for visualizing the characteristics of a univariate sample, as well as for visualizing the relationship between a continuous variable and a categorical variable. Despite its simplicity, many key distributional characteristics can be evaluated from the boxplot: namely, location, scale, skewness, and tails. This has made it popular among practitioners.

In addition, its simplicity is beneficial from a privacy standpoint. The boxplot only requires estimating a few summary statistics in order to convey the distributional characteristics of a univariate sample, making the boxplot efficient in its use of privacy budget. For instance, by contrast, a histogram can also be used to convey the same information. However, a histogram is generally a consistent estimate of the population density, which intuitively, should then require more noise to ensure privacy.

Motivated by these observations, we take the first steps in developing and studying differentially private boxplots. First, one can observe that a boxplot is made up of sample quantiles. It is then natural to first consider how well a boxplot constructed from naive applications of existing differentially private quantile algorithms. Succinctly:

---

[*]Equal contribution  [1]Department of Mathematics and Statistics, York University, Toronto, Canada. Correspondence to: Kelly Ramsay <kramsay2@yorku.ca>.

*Proceedings of the 42nd International Conference on Machine Learning*, Vancouver, Canada. PMLR 267, 2025. Copyright 2025 by the author(s).

**How well does the "naive" private boxplot perform?**

On this front, with a combination of new theoretical results and simulations, we demonstrate that this method may fail on data coming from common distribution families. We focus on the state-of-the-art algorithms `JointExp` (Gillenwater et al., 2021), `PrivateQuantile` (Smith, 2011), `ApproxQuantile` (Kaplan et al., 2022), and `unbounded` (Durfee, 2023). Due to limited space, our rationale for this choice, and a full literature review on private quantiles, including how our results build on this literature, is relegated to Appendix D. See also the related problem of differentially private range estimation (Kaplan et al., 2020). While range queries can be used to compute quantiles (Bun et al., 2015; Kulkarni, 2019; Kaplan et al., 2020), our focus is on more practical methods that estimate differentially private quantiles. In summary:

- We prove that `JointExp`, `PrivateQuantile` and `ApproxQuantile` are (almost always) inconsistent for extreme quantiles (Lemma 4.1), which makes them a poor choice for generating the whiskers in the box-plot. (To our knowledge, no results concerning the consistency of extreme private quantiles previously existed. Lalanne et al. (2023b) proved consistency for inner quantiles, under the assumption of a lower bound on the density, which would not apply to extreme quantiles.)

- We prove that `unbounded` is consistent for extreme quantiles (Lemma B.3) . This result also confirms an empirical observation of Durfee (2023), which says that the `unbounded` quantile algorithm performs better than existing algorithms for estimating extreme quantiles. (To our knowledge, no statistical results concerning `unbounded` previously existed.)

- Our simulations reveal that the `unbounded` algorithm does not perform as well as `JointExp`, `ApproxQuantile` or `PrivateQuantile` for estimating the inner quantiles of Gaussian data. This makes it a poor choice for generating the box in the plot. We support this finding by providing matching (up to logarithmic factors) upper and lower bounds for the sample complexity of `JointExp`, `ApproxQuantile` and `PrivateQuantile` for estimating inner quantiles, under general distributional assumptions (Theorem 4.2 and Theorem 4.4). In particular, we only assume that the density is positive in a neighborhood of the theoretical quantile and bounded everywhere (Condition 1). Previous concentration results for these algorithms assume the population density has bounded support (Lalanne et al., 2023a). The lower bound is new, and does not follow from the

lower bound on private medians (Tzamos & Vlatakis-Gkaragkounis, 2020).

Given that the naive boxplot is not satisfactory, we then sought to answer:

**Can we develop a novel private boxplot, which performs well at generating both the box and whiskers?**

To this end, building on the aforementioned results concerning private quantiles, we present a novel differentially private boxplot, which we call `DPBoxplot`. We then evaluate its ability to represent key features–location, scale, skewness, and tails–both theoretically and empirically. Figure 1 provides an informal, graphical illustration of our approach. Specifically, we use `JointExp` for estimating the quartiles and the median to construct the box, and `unbounded` for determining extreme values necessary for the whiskers and the number of outliers. Instead of revealing the precise locations of outliers, we privately disclose their count using the Laplace mechanism. In summary, our main contributions concerning the new differentially private boxplot are:

- We carefully combine private quantile estimators, ensuring that those used for the location and scale of the boxplot are estimated optimally (Theorem 4.2), while the extreme quantiles required to depict skewness and tails remain consistent (Theorem 4.5).

- Extensive simulations demonstrate that `DPBoxplot` is often more accurate than those constructed naively using existing differentially private quantile methods, see Section 5.

- We conduct a differentially private exploratory data analysis, showcasing the practical utility and shortcomings of `DPBoxplot`, see Section 6.

## 2. Preliminaries

First, we briefly review the boxplot. It is also helpful to introduce some notation used throughout the paper. We define a dataset of size $n \in \mathbb{N}$ to be a set of $n$ real numbers. Let $\mathcal{D}_n$ be the set of datasets of size $n$ and let $\mathbf{X}_n = \{X_1, \ldots, X_n\}$ be a dataset size $n$. In this work, we assume the analyst has access to a dataset, denoted by $\mathbf{X}_n$. We will also assume that $\mathbf{X}_n$ consists of $n$ independent draws of from a common population distribution or measure, denoted $\nu$. We consider the traditional boxplot, which consists of a box with a line drawn through it, with two whiskers emanating from the lower and upper bounds of the box, see Figure 1. The box is made up of the median and the quartiles, while the whiskers, represent the tails and skewness of the data. Points beyond the whiskers are explicitly plotted. Together, this 5 number summary describes the empirical distribution of the data,

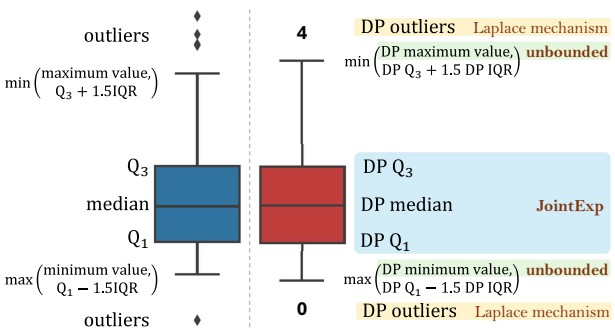

*Figure 1.* Graphical illustration of algorithms for the traditional non-private boxplot (left) and our proposed differentially private boxplot `DPBoxplot` (right).

including its location, scale, skewness and tails. A more detailed review of the boxplot can be seen in Appendix A.

## 2.1. Differential privacy

Next, we introduce differential privacy. First, we say that $\mathbf{Y}_n \in \mathcal{D}_n$, is adjacent to $\mathbf{X}_n$ if $\mathbf{X}_n$ and $\mathbf{Y}_n$ differ in exactly one point. Let $\mathcal{A}_n$ be the set of pairs of adjacent datasets of size $n$. Next, denote by $P_{\mathbf{X}_n}$ a probability measure which depends on $\mathbf{X}_n$. For a space $S$, let $\mathscr{B}(S)$ denote the Borel sets of $S$ and let $\mathcal{M}_1(S)$ be the set of Borel probability measures over $S$. Formally, $P_{\mathbf{X}_n}$ is a map from $\mathbb{R}^n$ to $\mathcal{M}_1(\mathbb{R}^d)$, for some $d \in \mathbb{N}$. We can now define differential privacy.

**Definition 2.1.** Given $\epsilon > 0$, the quantity $\theta \sim P_{\mathbf{X}_n}$ is $\epsilon$-differentially private if

$$\sup_{(\mathbf{X}_n, \mathbf{Y}_n) \in \mathcal{A}_n} \sup_{B \in \mathscr{B}(\mathbb{R}^d)} \frac{P_{\mathbf{X}_n}(B)}{P_{\mathbf{Y}_n}(B)} \le e^\epsilon. \tag{1}$$

Here, $\theta$ is a $d$-dimensional differentially private quantity and $\epsilon$ is the privacy budget, where values of $\epsilon$ which are closer to zero enforce stricter privacy constraints.

The proposed differentially private boxplot makes use of existing differentially private algorithms. Specifically, the Laplace mechanism, and algorithms for computing differentially private quantiles. The Laplace mechanism (Dwork et al., 2006) is a fundamental mechanism for constructing differentially private statistics. Let $Z$ be a standard Laplace random variable. Specifically, for a statistic $T: \mathcal{D}_n \to \mathbb{R}$, define its global sensitivity to be $\sup_{(\mathbf{X}_n, \mathbf{Y}_n) \in \mathcal{A}_n} |T(\mathbf{X}_n) - T(\mathbf{Y}_n)|$. If $T$ has global sensitivity bounded by 1, then $\tilde{T} = T(\mathbf{X}_n) + Z/\epsilon$ is a differentially private quantity. This is known as the Laplace mechanism.

## 2.2. Private quantiles

Of course, a private boxplot relies on differentially private quantiles. The proposed private boxplot relies on

the `unbounded` algorithm of Durfee (2023) and the `JointExp` algorithm of Gillenwater et al. (2021). These algorithms are the main focus of our theoretical exposition. However, the `PrivateQuantile` is a special case of `JointExp` and so our results also apply to `PrivateQuantile`. In addition, combining our results with those of Kaplan et al. (2022) allows our results to also hold for `ApproxQuantile`. We do, however, consider the performance of the naive boxplot constructed from all three of these algorithms in our simulation study, see Section 5.

We now give a brief overview of `JointExp` and `unbounded`. `JointExp` works as follows: Suppose we wish to estimate $m \in \mathbb{N}$ quantiles of levels $0 < q_1 < \ldots < q_m \le 1$. For an integer $k \in \mathbb{N}$, let $[k]$ denote the set $\{1, \ldots, k\}$. Define the $j$th quantile generated by `JointExp` for a given $j \in [m]$ to be $\tilde{\xi}_{q_j}$, and let $\tilde{\xi}_q = (\tilde{\xi}_{q_1}, \ldots, \tilde{\xi}_{q_m})$. For a given $\mu \in \mathcal{M}_1(\mathbb{R})$, let $F_\mu$ be the cumulative distribution function (CDF) of $\mu$ and, if it exists, let $f_\mu$ be its probability density function. For the dataset $\mathbf{X}_n$, let $\hat{\nu}$ be the corresponding empirical measure, so that $F_{\hat{\nu}}$ is then the associated empirical CDF. For $x \in \mathbb{R}^m$ with $x_1 < \ldots < x_m$, define the following utility function: $\phi(x) = -\sum_{j=1}^{m+1} |F_{\hat{\nu}}(x_j) - F_{\hat{\nu}}(x_{j-1}) - (q_j - q_{j-1})|$, where $x_0 = -\infty$, $x_{m+1} = \infty$, $q_0 = 0$ and $q_{m+1} = 1$. Then $\tilde{\xi}_q$ is a random vector drawn from the density $f_{Q_{\mathbf{X}_n}}(x) \propto \exp(-n\phi(x)/2\epsilon)\mathbb{1}\{x \in [a,b]^m\}$, where $Q_{\mathbf{X}_n}$ is used to denote the distribution of $\tilde{\xi}_q$, given $\mathbf{X}_n$.[1] It was shown by Gillenwater et al. (2021) that $\tilde{\xi}_q$ is $\epsilon$-differentially private. Furthermore, Lalanne et al. (2023b) showed that this algorithm is consistent if $\nu$ is continuous. If one suspects that $\nu$ contains atoms, then we recommend that one uses the jittering modification proposed by Lalanne et al. (2023b). We show in Section 4 that, for a large class of distributions, this algorithm has optimal sample complexity for estimating a single inner quantile, and therefore, for the scale and location of the boxplot. (Here we are only interested in estimating three quantiles, and so, how sample complexity scales in $m$ in not of particular concern.)

The original `unbounded` algorithm produces a private quantile given a lower bound on the data. We present a slightly modified version, which performs better for extreme quantiles. For a given quantile $q \in [1/2, 1]$ and $a > 0$, let $V_0, V_1, \ldots$ be independent, standard exponential random variables and let $\beta_n > 1$. Define $i^* = \inf\{i \in \mathbb{N}: F_{\hat{\nu}}(\beta_n^i + a - 1) + \frac{2}{n\epsilon}V_i \ge q + V_0\frac{2}{n\epsilon}\}$. Here $i^*$ is the smallest integer $i$ such that a noisy empirical CDF computed at $\beta_n^i + a - 1$ is greater than a noisy version of $q$. The quantile estimate generated by the `unbounded` algorithm is given by: $\tilde{\psi}_q = \beta_n^{i^*} + a - 1$. The intuition is that if a noisy version of $F_{\hat{\nu}}(\beta_n^{i^*} + a - 1)$ is close to a noisy version of $q$,

---
[1]Note $f_{Q_{\mathbf{X}_n}}$ is only defined for $x \in \mathbb{R}^m$ with $x_1 < \ldots < x_m$.

then $\beta_n^{i^*} + a - 1$ should be close to the nonprivate quantile. If $q \in (0, 1/2)$, then, we apply the above procedure to the dataset $-\mathbf{X}_n = \{-X_1, \ldots, -X_n\}$, with input parameters $1 - q + 1/n$ and $a = -b$. It follows from (Durfee, 2023) that this algorithm is $\epsilon$-differentially private. In the proposed differentially private boxplot algorithm, we use unbounded to estimate the minimum and maximum of the dataset. For a measure $\mu \in \mathcal{M}_1(\mathbb{R})$, let $\xi_{p,\mu} = \inf\{x \in \mathbb{R} : p \leq F_\mu(x)\}$ be the associated $p$th quantile, $\min(\mu) = \inf\{x : F_\mu(x) > 0\}$ and $\max(\mu) = \inf\{x : F_\nu(x) = 1\}$. We show that if $\mathbf{X}_n$ is an independent sample from $\nu$, then $\tilde{\psi}_1$ is weakly consistent for $\max(\nu)$, and $\tilde{\psi}_{1/n}$ is weakly consistent for $\min(\nu)$, see Lemma B.3. By contrast, we show that when the support of $f_\nu$ is bounded below, JointExp is inconsistent for $\min(\nu)$ when $q = 1/n$, unless $a = \min(\nu)$, see Lemma 4.1. Lastly, we note that instead of exponential noise, one could also use Laplace or Gumbel noise (Durfee, 2023). We found this to make little difference in simulation, and so we only present the version which incorporates exponential noise.

## 3. A differentially private boxplot

We can now introduce the differentially private boxplot. Given lower and upper bounds on the data $a$ and $b$, we first generate the minimum and maximum estimates using the unbounded algorithm, $\tilde{\psi}_{1/n}$ and $\tilde{\psi}_1$ with a privacy budget of $3\epsilon/16$ each. If one does not wish to supply input bounds for the data, one can use the "fully unbounded" version of unbounded (Durfee, 2023). However, in simulation, (see Section 5), we have observed that the procedure is still accurate, even when the input bounds are very loose. Critically, we use the unbounded algorithm because JointExp (and, by consequence, PrivateQuantile and ApproxQuantile) is often inconsistent for estimating the aforementioned extreme quantiles, unless the input bounds $a$ and $b$ match $\min(\nu)$ and $\max(\nu)$, respectively, see Lemma 4.1. Next, we run one instance of JointExp with a total privacy budget of $\epsilon/2$ to generate $\tilde{\xi}_{1/4}, \tilde{\xi}_{1/2}, \tilde{\xi}_{3/4}$ simultaneously to be the lower box bound, center line and upper box bound, respectively. We then calculate the private interquartile range $\tilde{\text{IQR}} = \tilde{\xi}_{3/4} - \tilde{\xi}_{1/4}$. Let $\tilde{\ell}_1 = \tilde{\xi}_{1/4} - 1.5 \times \tilde{\text{IQR}}$, $\tilde{u}_1 = \tilde{\xi}_{3/4} + 1.5 \times \tilde{\text{IQR}}$ and $\lambda_n > 0$ be the buffer parameter. The lower and upper whisker are defined as

$$\tilde{\ell} = \begin{cases} \tilde{\ell}_1 & \tilde{\xi}_{1/n} \leq \lambda_n |\tilde{\ell}_1| + \tilde{\ell}_1 \\ \tilde{\psi}_{1/n} & \tilde{\xi}_{1/n} > \lambda_n |\tilde{\ell}_1| + \tilde{\ell}_1, \end{cases}$$

and

$$\tilde{u} = \begin{cases} \tilde{u}_1 & \tilde{\xi}_1 \geq \tilde{u}_1 - \lambda_n |\tilde{u}_1| \\ \tilde{\psi}_1 & \tilde{\xi}_1 < \tilde{u}_1 - \lambda_n |\tilde{u}_1| \end{cases},$$

respectively. To elaborate, the role of the minimum and the maximum of the dataset in a traditional boxplot is played

by the estimated extreme quantiles $\tilde{\psi}_{1/n}$ and $\tilde{\psi}_1$. Given that estimated extreme quantiles are more variable than the estimated inner quantiles, we add a buffer $\lambda_n$ to account for this. Specifically, we take the upper whisker equal to the extreme quantile $\tilde{\psi}_1$ when $\tilde{\psi}_1$ is $(\lambda_n \times 100)\%$ smaller than 1.5 times the $\tilde{\text{IQR}}$. An analogous procedure is done for the lower whisker. The role of $\lambda_n$ is to account for the fact that the algorithm is more prone to erroneously replacing the IQR-based whisker with an extreme quantile. This is because extreme quantiles are more variable than the inner quantiles. To mitigate this, we introduce $\lambda_n$ as a trust parameter that favors the IQR whisker—allowing it to be replaced only when the extreme quantile is smaller in magnitude by at least $(\lambda_n \times 100)\%$. We take $\lambda_n = n^{-1/4}$. Simulation results (Appendix C.2) indicate that the performance of the method is not overly sensitive to the choice of this parameter.

The boxplot includes the observations that lie above and below the upper and lower whiskers, respectively. We cannot release such data points under the constraints of differential privacy. Instead, we plot a noisy version of the number of points above $\tilde{u}$, denoted $\tilde{o}_u$ and below $\tilde{\ell}$, denoted $\tilde{o}_\ell$. These are generated via the Laplace mechanism, where it is easy to see that the count of observations above or below a threshold has global sensitivity 1. Lastly, we attribute less privacy budget for computing the outliers, as we deem these values to be of less interest than the box itself. We assign each noisy outlyingness number a privacy budget of $\epsilon/16$. Together, these seven values make up the differentially private boxplot: $\tilde{B}(\mathbf{X}_n, \epsilon) = \tilde{B}(\hat{\nu}, \epsilon) = (\tilde{o}_\ell, \tilde{\ell}, \tilde{\xi}_{1/4}, \tilde{\xi}_{1/2}, \tilde{\xi}_{3/4}, \tilde{u}, \tilde{o}_u)$. The algorithm, which we call DPBoxplot, is summarized in Algorithm 3, see also, Figure 1. It follows from sequential composition that DPBoxplot is $\epsilon$-differentially private.

Note that the time and space complexities of DPBoxplot are given by the maximum of those of the differentially private quantile algorithms unbounded and JointExp. Since the unbounded quantile can be computed in linear time (Durfee, 2023), the overall complexity of DPBoxplot is determined by that of JointExp, which is $O(n \log(n))$ (Gillenwater et al., 2021).

## 4. Theoretical results

In this section, we derive several results concerning private quantiles and the different elements of the private boxplot. Recall that we assume that the dataset $\mathbf{X}_n$ consists of $n$ independent, identically distributed random variables drawn from a population measure $\nu \in \mathcal{M}_1(\mathbb{R})$. We first present a lemma which says that JointExp is inconsistent for extreme quantiles. We then present an upper bound on the sample complexity of the inner quantiles generated from JointExp. After which, we present a minimax lower bound for privately estimating a quantile from a population measure lying in a general set, which matches the upper

**Algorithm 1** `DPBoxplot`
> **Input:** data $\mathbf{X}_n, \epsilon, a, b, \lambda_n$
> {Construct the private quantiles[2]}
> $\tilde{\psi}_1 \leftarrow$ unbounded$(1, a, b, 3\epsilon/16)$
> $\tilde{\psi}_{1/n} \leftarrow$ unbounded$(1/n, a, b, 3\epsilon/16)$
> $\tilde{\xi}_q \leftarrow$ JointExp$(q, a, b, \epsilon/2)$
> {Ensure the box contains the median}
> $\tilde{\xi}_{1/4} \leftarrow \min(\tilde{\xi}_{1/4}, \tilde{\xi}_{1/2})$
> $\tilde{\xi}_{3/4} \leftarrow \max(\tilde{\xi}_{3/4}, \tilde{\xi}_{1/2})$
> {Construct the whiskers and outlyingness counts}
> $\tilde{\ell} \leftarrow \tilde{\xi}_{1/4} - 1.5(\tilde{\xi}_{3/4} - \tilde{\xi}_{1/4})$
> $\tilde{u} \leftarrow \tilde{\xi}_{3/4} + 1.5(\tilde{\xi}_{3/4} - \tilde{\xi}_{1/4})$
> **if** $\tilde{\psi}_{1/n} > \lambda_n |\tilde{\ell}| + \tilde{\ell}$ **then**
> $\quad \tilde{\ell} \leftarrow \tilde{\psi}_{1/n}$
> $\quad \tilde{o}_\ell \leftarrow 0$
> **else**
> $\quad \tilde{o}_\ell \leftarrow \sum_{i=1}^n \mathbb{1}\left\{x_i < \tilde{\ell}\right\} + \text{Laplace}(0, \frac{1}{\epsilon/16})$
> **end if**
> **if** $\tilde{\psi}_1 < \tilde{u} - \lambda_n |\tilde{u}|$ **then**
> $\quad \tilde{u} \leftarrow \tilde{\psi}_1$
> $\quad \tilde{o}_u \leftarrow 0$
> **else**
> $\quad \tilde{o}_u \leftarrow \sum_{i=1}^n \mathbb{1}\left\{x_i > \tilde{u}\right\} + \text{Laplace}(0, \frac{1}{\epsilon/16})$
> **end if**
> **Output:** $\tilde{B}(\mathbf{X}_n, \epsilon) = (\tilde{o}_\ell, \tilde{\ell}, \tilde{\xi}_{1/4}, \tilde{\xi}_{1/2}, \tilde{\xi}_{3/4}, \tilde{u}, \tilde{o}_u)$

bound up to logarithmic terms. Lastly, we present a result that says the whiskers and outlyingness numbers are consistent for their population counterparts.

We now define a boxplot rigorously, as a function of a measure on the set of real numbers, which is convenient mathematically. The non-private boxplot constructed from the data is taken to be the boxplot computed on the empirical measure of the dataset $\mathbf{X}_n$, denoted by $\hat{\nu}$. For $\nu \in \mathcal{M}_1(\mathbb{R})$, let $\text{IQR}(\nu) = \xi_{3/4,\nu} - \xi_{1/4,\nu}$. Now, letting $\ell_{1,\nu} = \xi_{1/4,\nu} - 1.5 \times \text{IQR}(\nu)$ and $u_{1,\nu} = \xi_{3/4,\nu} + 1.5 \times \text{IQR}(\nu)$, the whiskers are defined as $\ell_\nu = \max(\ell_{1,\nu}, \min(\nu))$ and $u_\nu = \min(u_{1,\nu}, \max(\nu))$. Lastly, we can define $o_{\ell,\nu} = F_\nu(\ell_\nu) - \nu(\{X = \ell_\nu\})$, $o_{u,\nu} = 1 - F_\nu(u_\nu)$. The "population" boxplot is the following seven number summary $B(\nu) = (o_{\ell,\nu}, \ell_\nu, \xi_{1/4,\nu}, \xi_{1/2,\nu}, \xi_{3/4,\nu}, u_\nu, o_{u,\nu})$.

The next lemma says that JointExp (and, by consequence, PrivateQuantile and ApproxQuantile since these algorithms are equivalent for $m = 1$ quantile) is inconsistent for the minimum of $\nu$ when we set $q = O(1/n)$ if the input bounds do not exactly match those of the support of the population distribution, and the support of the population distribution is bounded.

**Lemma 4.1.** *For $-\infty < a < b < \infty$, $0 < q \le 1$, if there exists $M > a$ such that $\nu(\{X \le M\}) = 0$, then for any*

$0 < t < M - a$, *we have that*

$$\Pr\left(|\tilde{\xi}_q - \xi_{q,\nu}| \ge t\right) \ge e^{-\epsilon \frac{nq}{2}} \frac{M - t - a}{b - a}.$$

In this case, if $q = n^{-1}$, then the sample complexity is bounded below by infinity, which implies inconsistency. The proof of Lemma 4.1 can be seen in Appendix B.

Before getting to our next results, we introduce a condition on the population distribution. For $L, r > 0$ and $q \in (0, 1]$, let $\mathcal{G}_{L,r,q}$ be the set of absolutely continuous measures $\mu \in \mathcal{M}_1(\mathbb{R})$ such that $\inf_{\xi_q - r \le x \le \xi_q + r} f_\mu(x) \ge L > 0$. Therefore, if the population measure $\nu \in \mathcal{G}_{L,r,q}$, then it has a density which is bounded below by $L$ in a neighborhood of size $r$ around its $q$th quantile.

*Condition* 1. Given $p = (p_1, \ldots, p_m)$ with $0 < p_1 < \ldots < p_m \le 1$ then $a \le \xi_{p_1} < \xi_{p_m} \le b$ and there exists $K, r, L > 0$ such that $\nu \in \cap_{i=1}^m G_{L,r,p_i}$ and $\sup_{x \in [a,b]} f_\nu(x) \le K$.

This condition has three requirements. First, we require that the interval $[a, b]$ contains the population quantiles we wish to estimate. Our theoretical and simulation results show that this interval can be chosen loosely, as the error in our estimates does not depend strongly on the size of this interval. Thus, this is not a strict requirement. Next, we require that for each $p_i$, $i \in [m]$, in a neighborhood of size $r$ of the $p_i$th quantile of $\nu$, the density is bounded away from 0. This requirement is standard in quantile estimation, e.g, (Tzamos & Vlatakis-Gkaragkounis, 2020; Lalanne et al., 2023a). Lastly, we require that the population density is bounded above everywhere. Requiring the density being bounded above is a weak restriction, which is satisfied by many distribution families, such as the Gaussian, Beta and Gamma families.

We can now state an upper bound on the sample complexity of the private quantiles generated via JointExp, and by consequence, PrivateQuantile and ApproxQuantile, see Remark 4.3. For $q = (q_1, \ldots, q_m)$ such that $0 < q_1 < \ldots < q_m \le 1$, define $\xi_{q,\nu} = (\xi_{q_1,\nu}, \ldots, \xi_{q_m,\nu})$. Next, for $x, y \in \mathbb{R}$, we write $x \vee y$ $(x \wedge y)$ for the maximum (minimum) of $x$ and $y$.

**Theorem 4.2.** *Given $-\infty < a < b < \infty$ and $q = (q_1, \ldots, q_m)$ such that $0 < q_1 < \ldots < q_m \le 1$, if Condition 1 holds with $p = q$, then there exists a universal constant $C > 0$ such that for all $0 < \gamma < 1$ and $0 < t \le r$, it holds that $\|\xi_{q,\tilde{\nu}} - \xi_{q,\nu}\| \le t$ with probability $1 - \gamma$, provided that*

$$n \ge C\left(m^{5/2} \frac{\log(1/\gamma) \vee m\log(m/ctL)}{t^2 L^2}\right.$$

$$\left. \bigvee m^2 \frac{\log(1/\gamma) \vee \log\left(\frac{K(b-a)}{q_1 \wedge (1-q_m) \wedge tL}\right)}{tL\epsilon}\right).$$

The proof of Theorem 4.2 relies on a general bound for the exponential mechanism detailed by Ramsay et al. (2024), it can be seen in Appendix B. Theorem 4.2 gives an upper bound on the sample complexity of the quantiles generated by JointExp for distributions who satisfy Condition 1 with $p$ being the vector of quantile levels to be estimated. Comparing to existing results, Lalanne et al. (2023a) give a concentration result for JointExp which yields the same upper bound given by Theorem 4.2, when the support of $f_\nu$ is bounded. Therefore, our bound is essentially a generalization of theirs, though we assume that $f_\nu$ is bounded above. The sample complexity of the median of Tzamos & Vlatakis-Gkaragkounis (2020), which is a different estimator, also matches the upper bound given in Theorem 4.2, under similar assumptions. Lastly, we note that by combining our techniques used to prove the upper bound given in Theorem 4.2 with the ApproxQuantile algorithm of Kaplan et al. (2022), one can obtain logarithmic scaling in $m$. However, in this context, $m = 3$, and this is not particularly relevant, thought may be interesting for other applications. In simulation, we observe almost identical performance between ApproxQuantile and JointExp for the purposes of DPBoxplot.

*Remark* 4.3. We now expand on how Theorem 4.2 applies to PrivateQuantile and ApproxQuantile. The PrivateQuantile algorithm only generates one quantile. So, for PrivateQuantile, we would be applying it $m$ times to estimate $m$ quantiles. Given that all three algorithms are the same with $m = 1$, Theorem 4.2 applies to PrivateQuantile with $m = 1$. Applying Theorem 4.2 $m$ times with $t = t/m^{1/2}$ yields the same bound that is given in Theorem 4.2. For ApproxQuantile, if you inspect the ApproxQuantile algorithm, it is made up of successive applications of PrivateQuantile at each level of a tree, with the input bounds depending on the previous levels of the tree. In that case, you can combine our bound for PrivateQuantile with $m = 1$, $t = t/(\log(m) + 1)$ and $\gamma = \gamma/m$. Then you can apply Lemma 3.2 of Kaplan et al. (2022) (Note their notation uses $\beta$ for $\gamma$ and $gamma$ for $t$) and the result follows immediately.

Next, we present a minimax lower bound for estimating a single quantile subject to differential privacy. Next, for a set $S$, let $\tilde{\mathcal{M}}_{1,\epsilon,n}(S)$ be the set of maps from $\mathcal{D}_n$ to $\mathcal{M}_1(S)$ that satisfy (1). Note that $\tilde{\mathcal{M}}_{1,\epsilon,n}(S)$ is just a formal way of writing the set of all differentially private mechanisms whose output lies in $S$. Next, we write $T_\epsilon \sim \tilde{\mathcal{M}}_{1,\epsilon,n}(\mathbb{R})$ to denote the set of all univariate differentially private estimators.[3] For $0 < q \leq 1$, let $h_q(x) = x\sqrt{(-\log x - [\Phi^{-1}(q)]^2/2)/\pi}$ and $C_q = \operatorname{argmax}_{x>0} h_q(x)$. Denote the minimax risk for

---

[3]To be clear, given an element of $\tilde{\mathcal{M}}_{1,\epsilon,n}(\mathbb{R})$, say $P_.$, $T_\epsilon$ could be the associated estimator. Given a dataset $\mathbf{X}_n$ with empirical measure $\hat{\nu}$, $T_\epsilon(\hat{\nu})$ would be a draw from $P_{\mathbf{X}_n}$, or $T_\epsilon(\hat{\nu}) \sim P_{\mathbf{X}_n}$.

differentially private quantile estimation by $\mathcal{R}(\epsilon, L, r, q) = \inf_{T_\epsilon \sim \tilde{\mathcal{M}}_{1,\epsilon,n}(\mathbb{R})} \sup_{\nu \in \mathcal{G}_{L,r,q}} \mathbb{E}|T_\epsilon(\hat{\nu}) - \xi_{q,\nu}|.$

**Theorem 4.4.** *For all $n \geq 1$, $q \in (0, 1)$, $L, r > 0$ such that $rL \leq \sup_{x>0} h_q(x)$, it holds that*

$$n = \Omega\left(\frac{C_q^2}{L^2 t^2} \vee \frac{C_q}{Lt\epsilon}\right), \qquad (2)$$

*samples are required for $\mathcal{R}(\epsilon, L, r, q) \leq t$ to hold.*

Theorem 4.4 gives a lower bound on the sample complexity for estimating the $q$th quantile from a distribution whose density is bounded below by $L$ in a neighborhood of size $r$ around the $q$th population quantile. That is, every differentially privacy quantile estimator requires at least $\Omega\left(\frac{C_q^2}{L^2 t^2} \vee \frac{C_q}{Lt\epsilon}\right)$ samples in order to have a minimax risk bounded above by $t$. What is important for this context, is that Theorem 4.4 implies a lower bound on the sample complexity for estimating $m$ quantiles from a distribution whose density is bounded below by $L$ in neighborhoods of size $r$ around each of the $m$ population quantiles. That is, for all $q = (q_1, \ldots, q_m)$, $n \geq 1$, $L, r > 0$ with $rL \leq \inf_{j\in[m]} C_{q_j}$, it holds that (2) samples are also required for $\inf_{T_\epsilon \sim \tilde{\mathcal{M}}_{1,\epsilon,n}(\mathbb{R}^m)} \sup_{\nu \in \mathcal{G}_{L,r,q}} \mathbb{E}\|T_\epsilon(\hat{\nu}) - \xi_{q,\nu}\| \leq t$. Applying this lower bound in conjunction with Theorem 4.2 yields that the scale and location of the proposed private boxplot are estimated optimally, up to logarithmic factors. Note that Tzamos & Vlatakis-Gkaragkounis (2020) give a similar minimax lower bound on differentially private median estimation. However, their proof does not directly extend to estimating an arbitrary quantile, since it relies on a set of uniform distributions that have the property that the median coincides with the mean. The proof of Theorem 4.4 makes use of the differentially private version of Fano's inequality (Acharya et al., 2021), it is deferred to Appendix B.

Next, we show that the whiskers and the outlyingness numbers, when appropriately normalized, are weakly consistent for their population counterparts.

**Theorem 4.5.** *For $-\infty < a < b < \infty$, if $\lambda_n \to 0$ and $\beta_n \to 1$ as $n \to \infty$, and Condition 1 holds for $p = (1/4, 1/2, 3/4)$, then it holds that $\tilde{\ell} \xrightarrow{p} \ell_\nu$, $\tilde{u} \xrightarrow{p} u_\nu$ $\tilde{o}_\ell/n \xrightarrow{p} o_{\ell,\nu}$ and $\tilde{o}_u/n \xrightarrow{p} o_{u,\nu}$.*

Here $\xrightarrow{p}$ denotes convergence in probability. Theorem 4.5 says if the population density is bounded below in neighborhoods of each of the population median and the population quartiles, the population density is bounded above everywhere, and $[a, b]$ contain the quartiles, then the whiskers and outlyingness numbers will be consistent. To our knowledge, this is the first result concerning extreme, private quantiles. Together, Theorems 4.2 and 4.5 imply that, given a large enough sample size, DPBoxplot will correctly represent

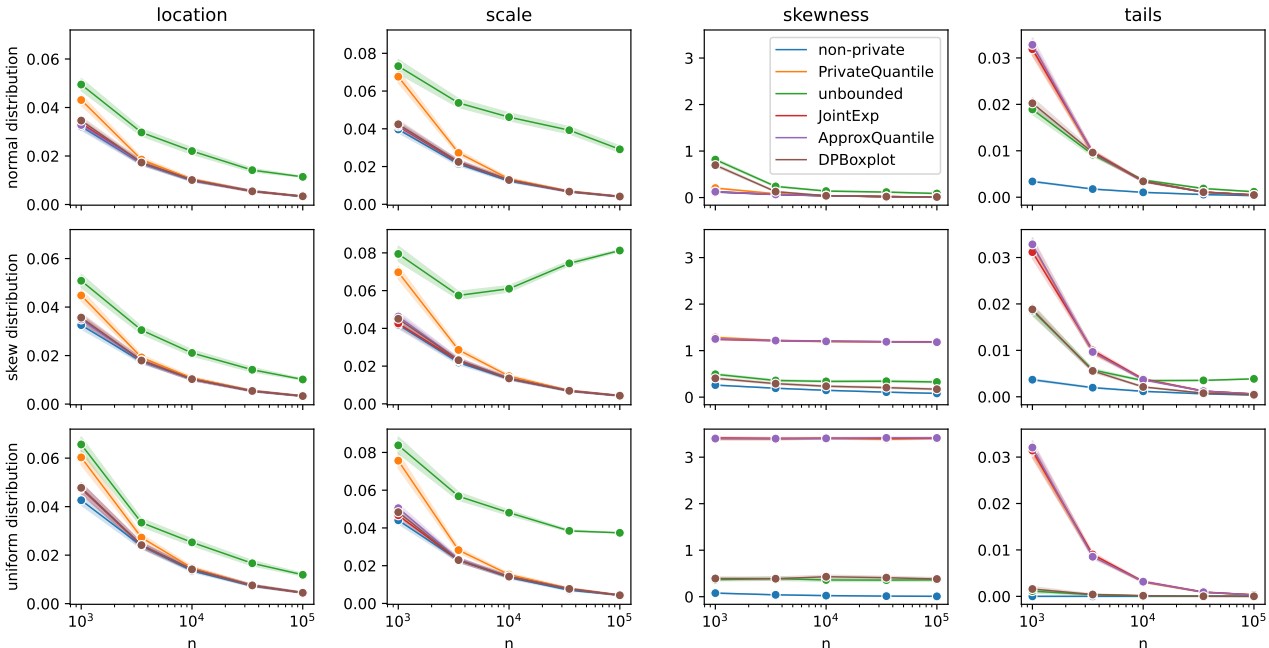

*Figure 2.* Average error in each metric with 95% confidence intervals between the different differentially private boxplots and their corresponding population boxplot (y-axis) with $\epsilon = 1$ and increasing $n$ (x-axis). The `DPBoxplot` algorithm is presented, along with the naive method paired with each of the algorithms `JointExp`, `ApproxQuantile`, `unbounded` and `PrivateQuantile` (line color). The different generated distributions are represented row-wise, and the different metrics: location, scale, skewness and tails are presented column-wise. The `non-private` line is the distance between the non-private boxplot and its corresponding population boxplot.

the location, scale, skewness and tail of the underlying distribution. Thus, making it statistically valid for one to use the private boxplot to describe samples in a differentially private exploratory data analysis.

## 5. Simulations

We now assess the empirical performance of `DPBoxplot`. We compare `DPBoxplot` to the "naive private boxplot" and the `non-private` sample boxplot. The naive private boxplot is one where a single differentially private quantile algorithm is used to generate all quantiles that are used in the boxplot. Using these quantiles, the whiskers and outlyingness counts are then constructed in the same manner as that of Algorithm 3. We consider the following quantile estimation methods: `PrivateQuantile`, `unbounded`, `ApproxQuantile` and `JointExp`. For all algorithms, we set $a = -50$ and $b = 50$ and $\lambda_n = n^{-1/4}$. We ran the same simulations with other values of $\lambda_n$ and found it did not alter the conclusions of the study, see Appendix C.2. For the `unbounded` algorithm, $\beta$ was set to the default value of 1.001 for both the naive boxplot and `DPBoxplot`.

We assess the boxplots across the four key metrics: scale, location, skewness and tails. We consider the error between each boxplot and the population boxplot. For `DPBoxplot`, errors in each of the components are quantified as follows:

$|\tilde{\xi}_{1/2} - \xi_{1/2,\nu}|$ (location), $|\tilde{\text{IQR}} - \text{IQR}(\nu)|$ (scale), $|\tilde{\ell} - \ell_\nu| + |\tilde{u} - u_\nu|$ (skewness), $|\tilde{o}_\ell - no_{\ell,\nu}| + |\tilde{o}_u - no_{u,\nu}|$ (tails). Errors for remaining boxplots are defined analogously.

Data were generated from the five distinct distributions parametrized to have mean 0 and variance 1: normal distribution (`normal`), skew normal distribution (`skew`), uniform distribution (`uniform`), beta distribution (`beta`), and an empirical distribution of 2019 NY airbnb listing prices (`airbnb`)[4]. We considered values of $n$ and $\epsilon$ and each scenario was simulated 1000 times. (For more details, see Appendix C.1). Results from `beta` and `airbnb` distributions, and those with $\epsilon = 0.5$ or $\epsilon = 5$ did not add additional insights, so we also defer to Appendix C.1. We also assessed `DPBoxplot`'s ability to compare multiple distributions simultaneously, concluding that `DPBoxplot` also performs well under this setting. For brevity, these results are deferred to Appendix C.3.

Figure 2 provides a comprehensive summary of the simulation results. Each column of figures corresponds to a distinct boxplot component, while each row pertains to a different generating distribution. The sample size is depicted along the $x$-axis, while the average error is represented on the $y$-axis. The line color within the figures denotes the method used to generate the boxplot.

---

[4]These data were sampled without replacement.

Consistent with Theorems 4.2, 4.4, and 4.5, the error for all components of `DPBoxplot` is converging to zero as $n$ increases in all scenarios. Further, we see that the error for `DPBoxplot` is similar to that of the error for `non-private`. This means that the sampling error is larger than that of the error attributed to privatization.

On the other hand, the naive boxplots do not always exhibit the same behavior. In particular, the inconsistency of the whiskers for the naive methods based on `JointExp`, `ApproxQuantile`, and `PrivateQuantile` is reflected through poor performance in the skew metric for the `skew` and `uniform` distributions. On the other hand, though it does well at estimating the extreme quantiles, the `unbounded` algorithm worse than the other algorithms for estimating scale and location. We see that `DPBoxplot` retains the best of both worlds.

In the setting of normally distributed data at small sample sizes, `DPBoxplot` does perform worse than the naive methods in the skewness metric. This happens because at small sample sizes, the `unbounded` algorithm tends to underestimate the magnitude of the minimum and maximum of the dataset, while the other algorithms overestimate the magnitude of the minimum and maximum. Nevertheless, we can still conclude that overall, `DPBoxplot` exhibits consistently better behavior than the naive methods.

## 6. Case study

We now conduct an exploratory data analysis via boxplots, within the framework of differential privacy.

We consider two business inquiries, each with a privacy budget of $\epsilon = 1$. Each inquiry consists of several visualizations. The privacy budget for each visualization is proportional to the number of boxplots in the visualization. To elaborate, out of the total budget ($\epsilon = 1$), each visualization (and therefore, as a result of parallel composition, each boxplot) is assigned a privacy budget equal to the number of boxplots in the visualization, divided by the number of boxplots on all generated visualizations. We chose this allocation so that more budget is allotted to visualizations where the data is partitioned more times. The intuition is that if the data is partitioned more times, then each boxplot in the visualization has a sample size which is smaller. Code for replicating our visualizations has been made publicly accessible ().

We analyze a dataset containing Airbnb listing prices and associated metrics within New York City (NYC) in 2019 (Kaggle, 2019). After removing listings priced above 500 US dollars (USD) and requiring minimum nights of stay fewer than 10, this dataset has $n = 40738$ observations and $d = 4$ explanatory variables of business interest.

We only consider listings priced below 500 USD, and so we

set $a = 0$ and $b = 500$. We address two distinct business inquiries.

**Inquiry 1: Do discernible patterns emerge in Airbnb listing prices across various boroughs in New York City and differing room types?**

The dataset encompasses five distinct boroughs within New York City, namely the Bronx, Brooklyn, Manhattan, Queens, and Staten Island, alongside three offered room types: Entire Home, Private Room, and Shared Room. We present three visualizations: boxplots of prices by borough, room type, and by borough and room type combinations. These generate 5, 3, and 15 boxplots, respectively, totaling 23.

The differentially private boxplots are displayed in Figure 3, juxtaposed with non-private counterparts. Only looking at the differentially private boxplots, Figure 3 reveals that prices predominantly lie in the bottom end of the range [0,500], with a right skew and heavy right tail observed across all boroughs and rooms, except for in the Bronx, and for shared rooms. The distribution of prices in the Bronx still exhibits a right skew, but has light tails. The distribution of prices for shared rooms appears to be symmetric, with light tails. Notably, prices appear elevated for Manhattan. As expected, entire homes are priced higher than shared spaces. The right-most plot, which displays prices by both borough and room type, affirms previous observations, except shared rooms in Staten Island and the Bronx. Staten Island and the Bronx exhibits higher variability for shared rooms.

It is natural to compare the patterns observed on the differentially private boxplots to those observed in the non-private boxplots. As previously mentioned, Figure 3 juxtaposes the differentially private boxplots with the non-private boxplots. Many conclusions derived from the private visualizations persist in the non-private domain. However, there are several disparities, which are outlined are as follows: In reality, both the listing prices for shared rooms and properties in the Bronx do have a heavy, right tail. The second disparity occurs in the third plot, where non-private plots does not indicate higher prices and variability for properties in Staten Island and the Bronx in shared rooms. These discrepancies can be explained by small sample sizes and our conservative choice of privacy budget. For instance, the number of shared room listings in Staten Island is only 9. Overall, the patterns observed are generally consistent between the private and non-private boxplots. The principal disparities are attributable to sample size, a factor we encourage practitioners to consider when conducting differentially private exploratory analyses.

**Inquiry 2: Are there observable trends in Airbnb listing prices concerning minimum nights required for reservation and the types of rooms offered?**

We create a new categorical variable called minimum nights,

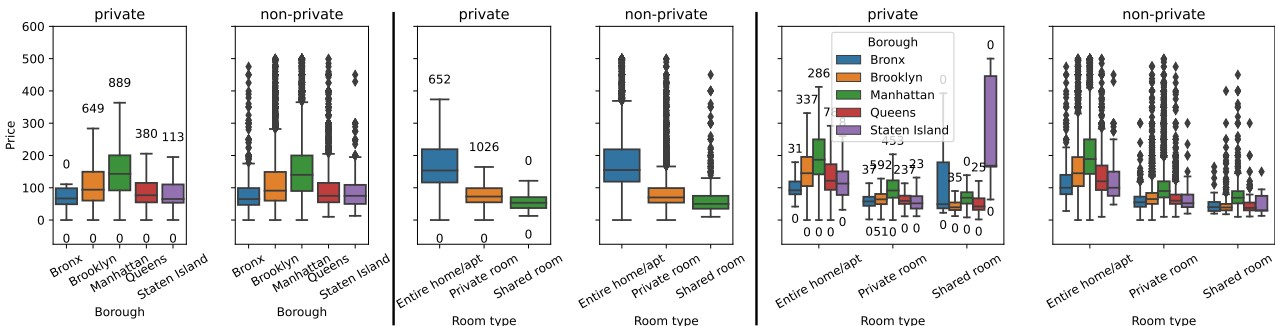

*Figure 3.* Private and non-private boxplots for inquiry 1. The boxplots display listing prices against borough, room type and room type separated by borough in columns 1, 2 and 3, respectively. Private boxplots are obtained with a total aggregated budget of $\epsilon = 1$.

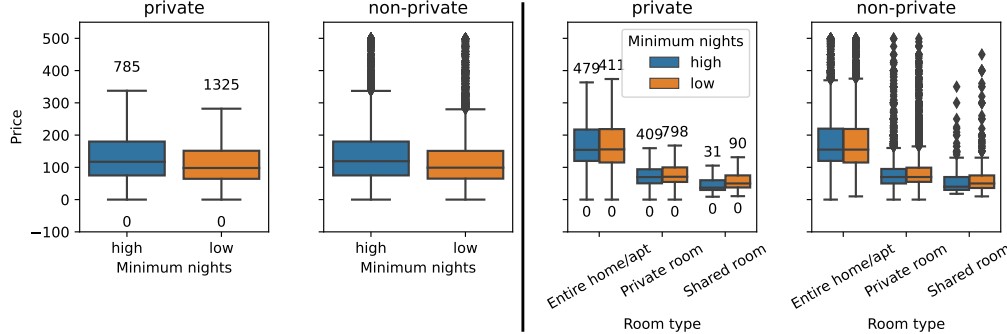

*Figure 4.* Private and non-private boxplots for inquiry 2. The boxplots display listing prices against minimum nights and room type separated by minimum nights in columns 1 and 2, respectively. Private boxplots are obtained with a total aggregated budget of $\epsilon = 1$.

in which a listing is assigned "low" if it has less than or equal to three minimum nights, and "high" otherwise. For this inquiry, we generate two visualizations: Boxplots of prices by minimum nights, and prices by combinations of minimum nights and room offered. These visualizations require 2, and 6 boxplots, respectively, totaling 8.

The differentially private boxplots are displayed in Figure 4, juxtaposed with their non-private counterparts. Again, we first analyze the patterns in the differentially private boxplots, and compare to those observed in the private boxplots afterward. The differentially private boxplots indicate the emergence of a phenomenon akin to Simpson's paradox. Specifically, a preliminary examination suggests that listings requiring a higher minimum number of nights are priced more steeply than their counterparts with lower minimum requirements. However, this trend disappears when the data is divided by room type. Listings for entire rooms exhibit no significant price differential based on minimum night requirement, while private rooms slightly favor lower minimum nights in terms of price. In shared rooms the median price does not seem to be significantly different.

A comparative analysis with non-private boxplots reaffirms these observations; we observe relatively consistent patterns between the private and non-private boxplots. The primary visual disparity pertains to the positioning of the lower whiskers on the boxplots. This underscores the recognized challenge associated with differentially private estimation of extreme quantiles. However, this discrepancy does not materially impede the analytical value of our findings.

## 7. Discussion

We have demonstrated that differentially private boxplots are a theoretically and practically viable standalone tool for private data analysis. Our work concerns differentially private data visualization, which, despite its impact potential, is severely underexplored. Though some work has been done in this area, a deep investigation into the practical aspects of differentially private exploratory data analysis is still needed, and a promising direction of future research. In particular, a large barrier for private exploratory analysis is optimal budget allocation for sequential and iterative workflows. Addressing this challenge is crucial to facilitate real-world adoption of differential privacy in data analysis pipelines.

**Code.** Official implementation is available at https://github.com/jairoadiazr/DPBoxplot.

## Acknowledgements

This work was supported by the Natural Sciences and Engineering Research Council of Canada (NSERC); through grants DGECR-2023-00311 and DGECR-2022-04531, respectively. We thank the anonymous reviewers and session chair for their valuable comments and suggestions, which helped improve the quality of this work.

## Impact statement

Our proposed methodology inherits the extensive implications, both beneficial and detrimental, of differential privacy. Differential privacy significantly enhances data confidentiality by introducing noise into datasets, which complicates the ability to link specific data points to individual identities. This greatly enhances the privacy awarded to individuals whose information is contained in such datasets. However, this method is not without its drawbacks. The primary challenge lies in the trade-off between privacy protection and data accuracy. The introduction of noise, while safeguarding privacy, may distort the data, potentially yielding inaccurate insights or conclusions. Such inaccuracies are especially problematic in contexts requiring high precision, such as policy development or clinical research, where exact data is crucial.

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

# A. A review of the boxplot

The boxplot consists of a box with a line drawn through it, with two whiskers emanating from the lower and upper bounds of the box. Specific points are sometimes indicated above or below the whiskers. The central line inside the box marks the median of the dataset. The box itself is constructed from the quartiles, detailing the middle 50% of $\mathbf{X}_n$. The lower (upper) whisker is constructed from the larger (smaller) of the minimum (maximum) of $\mathbf{X}_n$ and the lower (upper) quartile of $\mathbf{X}_n$ minus (plus) 1.5 times the interquartile range of $\mathbf{X}_n$. Lastly, any points in $\mathbf{X}_n$ falling outside of the whiskers are added to the plot. See Figure 3 for an example. The boxplot describes various characteristics of $\mathbf{X}_n$. The median line approximates the *location* of $\mathbf{X}_n$. The box itself approximates the spread or *scale* of $\mathbf{X}_n$. The *skewness* of $\mathbf{X}_n$ can be approximated as follows: A median line placed in the center of the box, along with whiskers that are similar in magnitude, suggests a symmetric distribution. A median line placed away from the center of the box or asymmetric whiskers indicates skewness. Additionally, the *tails* can be assessed by the number of points beyond the whiskers. Many such points signify outliers or heavy tails, highlighting data points that deviate significantly from the rest.

# B. Technical proofs

Here for $a, b \in \mathbb{R}$, $a \vee b$ ($a \wedge b$) denotes the maximum (minimum) of $a$ and $b$.

*Proof of Lemma 4.1.* We have that

$$\Pr\left(|\tilde{\xi}_{q_n} - \xi_{q_n,\nu}| \geq t\right) \geq \Pr\left(\tilde{\xi}_{q_n} \leq M - t\right) \geq \frac{\int_a^{M-t} e^{-\epsilon \frac{n|F_{\hat{\nu}}(x) - q_n|}{2}} dx}{\int_a^b e^{-\frac{n|F_{\hat{\nu}}(x) - q_n|}{2\epsilon}} dx} \geq e^{-\epsilon \frac{n q_n}{2}} \frac{M - t - a}{b - a}. \qquad \square$$

Before presenting the proof of Theorem 4.2, we first prove a general sample complexity bound for quantiles. For $t \geq 0$, $q = (q_1, \ldots, q_m)$ such that $0 < q_1 < \ldots < q_m < 1$, $-\infty < a < b < \infty$, and $\nu \in \mathcal{M}_1(\mathbb{R})$, define:

$$\alpha(t) = \inf_{a < x_1 < \ldots < x_m < b, \, \|x - \xi_{q,\nu}\| \geq t} \sum_{i=1}^{m+1} |q_j - q_{j-1} - F_\nu(x_j) + F_\nu(x_{j-1})|.$$

One may wish to recall that $\inf \emptyset = \infty$, and so $\alpha(t)$ is still defined if $\{a < x_1 < \ldots < x_m < b, \, \|x - \xi_{q,\nu}\| \geq t\} = \emptyset$.

**Lemma B.1.** *For $-\infty < a < b < \infty$, if $\nu \in \mathcal{M}_1(\mathbb{R})$ is absolutely continuous such that $\sup_{x \in [a,b]} f_\nu(x) \leq K < \infty$, then there exists universal constants $C, c > 0$ such that for all $q = (q_1, \ldots, q_m)$ such that $0 < q_1 < \ldots < q_m < 1$ and $a \leq \xi_{q_1,\nu} \leq \ldots \xi_{q_m,\nu} \leq b$, all $t \geq 0$ and $0 < \gamma < 1$, it holds that $\|\xi_{q,\tilde{\nu}} - \xi_{q,\nu}\| \leq t$, with probability $1 - \gamma$, provided that*

$$n \geq C m^2 \frac{\log(1/\gamma) \vee m \log m / c\alpha(t)}{\alpha^2(t)} \bigvee m \frac{\log(1/\gamma) \vee \log\left(\frac{K(b-a)}{q_1 \wedge (1-q_m) \wedge \alpha(t)}\right)}{\alpha(t)\epsilon}. \tag{3}$$

*Proof of Lemma B.1.* Given that $\tilde{\xi}_{q,\hat{\nu}}$ is a draw from the exponential mechanism, the result follows from an application of Corollary 7 of (Ramsay et al., 2024). Corollary 7 of (Ramsay et al., 2024) gives an upper bound on the sample complexity of a draw from the exponential mechanism, provided the utility function meets certain criteria. In order to apply Corollary 7 of (Ramsay et al., 2024), we must show that the following function

$$\phi'(x, \nu) = -\sum_{j=1}^{m+1} |F_\nu(x_j) - F_\nu(x_{j-1}) - (q_j - q_{j-1})| \mathbb{1}\{[x_1, x_m] \in [a, b]\} - \infty \cdot (1 - \mathbb{1}\{[x_1, x_m] \in [a, b]\}),$$

and $\nu$ satisfy three conditions. First, we must show that $\phi'(x, \nu)$ has a maximum, which is easily seen by definition, $\phi'(x, \nu)$ is maximized at $\xi_{q,\nu}$. The second requirement is that $\phi'(x, \nu)$ is $K$-Lipschitz, which follows by assumption. Lastly, Corollary 7 of (Ramsay et al., 2024) requires that $\phi'(x, \nu)$ is a $(L, \mathcal{F})$ regular function for some $L > 0$ with $\text{VC}(\mathcal{F}) < \infty$, see Definition E.2. It is easy to see that $\phi'(x, \nu)$ is $(m, \mathcal{F})$ regular function with $\text{VC}(\mathcal{F}) = 1$. In order to apply the bound given in Corollary 7 of (Ramsay et al., 2024), we must compute the discrepancy function of $\phi', \nu$, which is given by: $\alpha'(t) = \phi'(\xi_{q,\nu}, \nu) - \sup_{\|x - \xi_{q,\nu}\| \geq t} \phi'(x, \nu)$. A straightforward calculation yields that $\alpha'(t) = \alpha(t)$. Applying Corollary

7 of (Ramsay et al., 2024), yields that there exists a universal constant $C > 0$ such that for any $t > 0$ and $0 \leq \gamma \leq 1$, $\|\xi_{q,\tilde{\nu}} - \xi_{q,\nu}\| \leq t$ with probability at least $1 - \gamma$, if

$$n \geq C \left( m^2 \frac{\log(1/\gamma) \vee m \log m/c\alpha(t)}{\alpha^2(t)} \bigvee m \frac{\log\left(1/\gamma\right) \vee \log\left(\frac{(b-a)}{|\xi_{q_1,\nu}-a|\wedge|b-\xi_{q_m,\nu}|\wedge\alpha(t)/K}\right)}{\alpha(t)\epsilon} \right).$$

Define $a \gtrsim b$ $(a \lesssim b)$ if there is a universal constant $c > 0$ such that $a \geq cb$ $(a \leq cb)$. Next, using the Lipschitz assumption and the mean value theorem, we have that $|\xi_{q_1,\nu} - a| \wedge |b - \xi_{q_m,\nu}| \gtrsim [q_1 \wedge (1 - q_m)]/K$, which yields the desired result. $\square$

We can now prove Theorem 4.2.

*Proof of Theorem 4.2.* The proof is based on an application of Lemma B.1. All of the conditions of Lemma B.1 are satisfied by assumption, and so it remains to lower bound $\alpha$. To this end, let $k_x = \operatorname{argmax}_{i\in[m]} |x_i - \xi_{q_i,\nu}|$. Using this notation, for $0 \leq t \leq r$, the definition of $\alpha$ in conjunction with the mean value theorem yields that

$$\alpha(t) = \inf_{a<x_1<...<x_m<b,\, \|x-\xi_{q,\nu}\|\geq t} \sum_{j=1}^{m+1} |q_j - q_{j-1} - F_\nu(x_j) + F_\nu(x_{j-1})|$$

$$\geq \inf_{a<x_1<...<x_m<b,\, \|x-\xi_{q,\nu}\|\geq t} \sum_{j=1}^{k_x} |q_j - q_{j-1} - F_\nu(x_j) + F_\nu(x_{j-1})|$$

$$\geq \inf_{a<x_1<...<x_m<b,\, \|x-\xi_{q,\nu}\|\geq t} |\sum_{j=1}^{k_x} q_j - q_{j-1} - F_\nu(x_j) + F_\nu(x_{j-1})|$$

$$\geq \inf_{a<x_1<...<x_m<b,\, \|x-\xi_{q,\nu}\|\geq t} |q_{k_x} - F_\nu(x_{k_x})|$$

$$\geq tL/\sqrt{m}.$$

Applying this bound, in conjunction with Lemma B.1, yields the desired result. $\square$

Next, we prove Theorem 4.4.

*Proof of Theorem 4.4.* We apply Corollary 4 of (Acharya et al., 2021), of which a simpler version is restated below for clarity. For $\mathcal{P} \subset \mathcal{M}_1(\mathbb{R})$, let $n^* = \inf\{n \colon \inf_{T_\epsilon \sim \tilde{\mathcal{M}}_{1,\epsilon,n}(\mathbb{R})} \sup_{\nu\in\mathcal{P}} \int |T_\epsilon(\hat{\nu}) - T_\epsilon(\nu)| d\nu^n \leq \tau\}$.

**Corollary B.2** ((Acharya et al., 2021))**.** *For all $\epsilon, \tau > 0$ and any $\mathcal{P} \subset \mathcal{M}_1(\mathbb{R})$, let $\mathcal{Q} = \{Q_1, \ldots, Q_m\} \subset \mathcal{P}$. If for all $i \neq j$, it holds that $|T_\epsilon(Q_i) - T_\epsilon(Q_j)| \geq 3\tau$, $\mathrm{KL}(Q_i, Q_j) \leq \beta_n$, and $\mathrm{TV}(Q_i, Q_j) \leq \gamma$, then $n^* = \Omega\left(\log m(\beta_n^{-1} \vee (\gamma\epsilon)^{-1})\right)$.*

In order to apply Corollary B.2, we first define a class of Gaussian measures which lie in $\mathcal{G}_{L,r,q}$. Recall that we consider $\nu$ with densities which satisfy: $\inf_{x\in\xi_{q,\nu}\pm r} f_\nu(x) \geq L$. Let $P_{\mu,\sigma} = \mathcal{N}(\mu, \sigma^2)$ and let $h_{\mu,\sigma}$ denote the associated quantile function. We have that for $q \leq 1/2$, it holds that

$$\inf_{x\in h_{\mu,\sigma}(q)\pm r} f_{P_{\mu,\sigma}}(x) \geq (2\pi\sigma^2)^{-1/2} \exp\left(-[\operatorname{erf}^{-1}(2q-1) - r/\sqrt{2}\sigma]^2\right)$$

$$\geq (2\pi\sigma^2)^{-1/2} \exp\left(-r^2/2\sigma^2 - [\operatorname{erf}^{-1}(2q-1)]^2\right).$$

Taking $\sigma = C_q/\sqrt{2\pi}L$ gives that

$$\inf_{x\in h_{\mu,\sigma}(q)\pm r} f_{P_{\mu,\sigma}}(x) \geq L/C_q \exp\left(-[\operatorname{erf}^{-1}(2q-1)]^2 - \pi r^2 L^2/C_q^2\right).$$

Now, note that

$$\operatorname{erf}^{-1}(2q-1) = \Phi^{-1}(q)/\sqrt{2},$$

which in turn implies that

$$\inf_{x \in h_{\mu,\sigma}(q) \pm r} f(x) \geq L \exp\left(-[\Phi^{-1}(q)]^2/2 - r^2 L^2\right)/\sqrt{\pi}.$$

Next, we have that

$$L/C_q \exp\left(-[\Phi^{-1}(q)]^2/2 - \pi r^2 L^2/C_q^2\right) \geq L$$

$$\iff \exp\left(-[\Phi^{-1}(q)]^2/2 - \pi r^2 L^2/C_q^2\right) \geq C_q$$

$$\iff [\Phi^{-1}(q)]^2/2 + \pi r^2 L^2/C_q^2 \leq -\log C_q$$

$$\iff rL \leq C_q \sqrt{(-\log C_q - [\Phi^{-1}(q)]^2/2)/\pi}.$$

We must then have that $rL \leq C_q \sqrt{(-\log C_q - [\Phi^{-1}(q)]^2/2)/\pi}$, which holds by assumption. Define $\mathcal{F}_{L,r} = \{\mathcal{N}(\mu, C_q^2/2\pi L^2): \mu \in \mathbb{R}\}$. We find the values of $\beta_n, \gamma$ and $\tau$ for $\mathcal{F}_{L,r}$. Consider the values $\mu_0, \mu_1$, and for $i{=}1$ or $i = 0$, denote $Q_i = P_{\mu_i, C_q^2/2\pi L^2}$. Note that for any quantile $q$, we have that $|\xi_{q,Q_0} - \xi_{q,Q_1}| = |\mu_0 - \mu_1|$. That is, the distance between the quantiles is just the difference between the means. For some $\tau > 0$, take any $\mu_0, \mu_1$ such that $|\mu_0 - \mu_1| \geq 3\tau$. It follows that $\mathrm{KL}(Q_0, Q_1) \lesssim \tau^2 L^2/C_q^2$ and $\mathrm{TV}(Q_0, Q_1) = 2\Phi(c\tau L/C_q) - 1 \lesssim L\tau/C_q$. Therefore, applying Corollary B.2 gives that

$$n^* = \Omega\left(\frac{C_q^2}{L^2 \tau^2} \vee \frac{C_q}{L\tau\epsilon}\right). \qquad \square$$

Before proceeding with the proof of Theorem 4.5, we first prove that the `unbounded` algorithm estimates extreme quantiles consistently.

**Lemma B.3.** *For all non-increasing sequences $\beta_n \to 1$ and $q_n \to 0$, and all $\nu \in \mathcal{M}(\mathbb{R})$, it holds that*

*i  if $\min(\nu) < b$, then $\tilde{\psi}_{q_n} \xrightarrow{P} \min(\nu)$.*

*ii  if $\max(\nu) > a$, then $\tilde{\psi}_{1-q_n} \xrightarrow{P} \max(\nu)$.*

*Proof.* First, without loss of generality, take $a = 0$. Next, using the fact that $V_0$ is a standard exponential random variable.

$$\Pr\left(2V_0/n\epsilon > 1 - q_n\right) = e^{-n(1-q_n)\epsilon/2}. \tag{4}$$

In addition, letting $c_n = 1/\log n$, the Dvoretzky–Kiefer–Wolfowitz inequality yields that

$$\Pr\left(\sup_{x \in \mathbb{R}} |F_{\hat{\nu}}(x) - F_\nu(x)| > c_n\right) \leq 2e^{-2n/(\log n)^2}. \tag{5}$$

It suffices to show that for all $t > 0$, it holds that $\Pr\left(|\tilde{\psi}_{1-q_n} - \max(\nu)| > t\right) \to 0$ as $n \to \infty$. (A symmetric argument then implies that also, $\Pr\left(|\tilde{\psi}_{q_n} - \min(\nu)| > t\right) \to 0$ as $n \to \infty$.) Letting $k$ be the integer such that $\tilde{\psi}_{1-q_n} = \beta_n^k - 1$, we have that

$$\Pr\left(|\tilde{\psi}_{1-q_n} - \max(\nu)| > t\right) = \Pr\left(\max(\nu) - \beta_n^k - 1 > t\right) + \Pr\left(\beta_n^k - 1 - \max(\nu) > t\right),$$

for which we can write

$$\Pr\left(\max(\nu) - \beta_n^k - 1 > t\right) = \Pr\left(k < \frac{\log(\max(\nu) - t - 1)}{\log \beta_n}\right) := \Pr\left(k < R_{t,1}\right),$$

$$\Pr\left(\beta_n^k - 1 - \max(\nu) > t\right) = \Pr\left(k > \frac{\log(\max(\nu) + t + 1)}{\log \beta_n}\right) := \Pr\left(k > R_{t,2}\right).$$

It suffices to show that $\Pr\left(k < R_{t,1}\right), \Pr\left(k > R_{t,2}\right) \to 0$ as $n \to \infty$. To this end, first, note that if

$$\frac{\log(\max(\nu) + t + 1)}{\log \beta_n} - \frac{\log(\max(\nu) - t - 1)}{\log \beta_n} < 1,$$

then $\Pr\left(\{k < R_{t,1}\} \cup \{k > R_{t,2}\}\right) = 1$. However, we have that $\beta_n$ is decreasing to 1. This immediately implies that for all $t > 0$, there exists $n_0 > 1$ such that for all $n > n_0$, we have that

$$\frac{\log\left(\max(\nu) + t + 1\right)}{\log \beta_n} - \frac{\log\left(\max(\nu) - t - 1\right)}{\log \beta_n} > 1.$$

Next, utilizing (5), the fact that $V_0 > 0$ and the definition of $V_i$, we have that

$$\Pr\left(k < R_{t,1}\right) = 1 - \prod_{i=1}^{R_{t,1}} \Pr\left(F_{\hat{\nu}}(\beta_n^i - 1) + 2V_i/n\epsilon < 1 - q_n + 2V_0/n\epsilon\right)$$

$$\leq 1 - \prod_{i=1}^{R_{t,1}} \Pr\left(F_{\nu}(\beta_n^i - 1) + 2V_i/n\epsilon < 1 - q_n - c_n\right) + 2e^{-2n/(\log n)^2}$$

$$\leq 1 - \prod_{i=1}^{R_{t,1}} \left(1 - \exp\left(-n\epsilon(1 - q_n - c_n - F_{\nu}(\beta_n^i - 1))/2\right)\right) + 2e^{-2n/(\log n)^2}$$

$$:= 1 - I + 2e^{-2n/(\log n)^2}. \tag{6}$$

Now, using the fact that $(1 + x/n)^n \geq 1 + x$ for all $n \geq 1$ and $|x| \leq n$, we have that

$$I = \prod_{i=1}^{R_{t,1}} \left(1 - \exp\left(-n\epsilon(1 - q_n - c_n - F_{\nu}(\beta_n^i - 1))/2\right)\right)$$

$$\geq \left(1 - \exp\left(-n\epsilon(1 - q_n - c_n - F_{\nu}(\beta_n^{R_{t,1}} - 1))/2\right)\right)^{R_{t,1}}$$

$$\geq 1 - R_{t,1} \exp\left(-n\epsilon(1 - q_n - c_n - F_{\nu}(\beta_n^{R_{t,1}} - 1))/2\right).$$

Now, applying the preceding inequality in conjunction with (6) yields that

$$\Pr\left(k < R_{t,1}\right) \leq R_{t,1} \exp\left(-n\epsilon(1 - q_n - c_n - F_{\nu}(\beta_n^{R_{t,1}} - 1))/2\right) + 2e^{-2n/(\log n)^2} \to 0,$$

as $n \to \infty$. On the other hand, for the term $\Pr\left(k > R_{t,2}\right)$, utilizing (4) and (5), we have that

$$\Pr\left(k > R_{t,2}\right) \leq \prod_{i=1}^{R_{t,2}} \Pr\left(F_{\nu}(\beta_n^i - 1) + 2V_i/n\epsilon < 1 - q_n/2 + c_n\right) + 2e^{-2n/(\log n)^2} + e^{-n\epsilon(1-q_n)/2}$$

$$\leq \Pr\left(F_{\nu}(\beta_n^{R_{t,2}} - 1) + 2V_{R_{t,2}}/n\epsilon < 1 - q_n/2 + c_n\right) + 2e^{-2n/(\log n)^2} + e^{-n\epsilon(1-q_n)/2}$$

$$\leq \mathbb{1}\left\{1 - q_n/2 + c_n - F_{\nu}(\beta_n^{R_{t,2}} - 1) \geq 0\right\} + 2e^{-2n/(\log n)^2} + e^{-n\epsilon(1-q_n)/2} \to 0,$$

as $n \to \infty$. □

We can now prove Theorem 4.5.

*Proof of Theorem 4.5.* We first prove that the whiskers are consistent. Theorem 4.2 implies that $\tilde{\ell}_1 \xrightarrow{\text{P}} \ell_{1,\nu}$ and Lemma B.3 implies that $\tilde{\ell}_2 \xrightarrow{\text{P}} \min(\nu)$. Continuous mapping theorem and the fact that $\lambda_n \to 0$ as $n \to \infty$ yields that $\tilde{\ell} \xrightarrow{\text{P}} \ell_\nu$. The same argument applies to the upper whisker.

For the outlyingness number, first, we have that $|o_{\ell,\nu}/n - \tilde{o}_\ell| \leq |o_{\ell,\hat{\nu}} - \tilde{o}_\ell/n| + |o_{\ell,\nu} - o_{\ell,\hat{\nu}}| := I + II$. Next, the properties of the Laplace distribution give that $\Pr\left(|o_{\ell,\hat{\nu}} - \tilde{o}_\ell/n| \geq t\right) \lesssim e^{-n\epsilon t}$. Therefore $I \xrightarrow{\text{P}} 0$. Next, using the fact that $\sup_{x \in \mathbb{R}} f_\nu(x) \leq K$, we have that $F_\nu$ is $K$-Lipschitz. It follows that $|o_{\ell,\nu}/n - o_{\ell,\hat{\nu}}/n| \leq K|\ell_{\hat{\nu}} - \ell_\nu| + \sup_{x \in \mathbb{R}} |F_\nu(x) - F_{\hat{\nu}}(x)|$. Next, the Dvoretzky–Kiefer–Wolfowitz inequality yields that $\sup_{x \in \mathbb{R}} |F_\nu(x) - F_{\hat{\nu}}(x)| \xrightarrow{\text{P}} 0$ and the fact that the whiskers are consistent implies that $K|\ell_{\hat{\nu}} - \ell_\nu| \xrightarrow{\text{P}} 0$. The same argument can be made for the upper outlyingness number. □

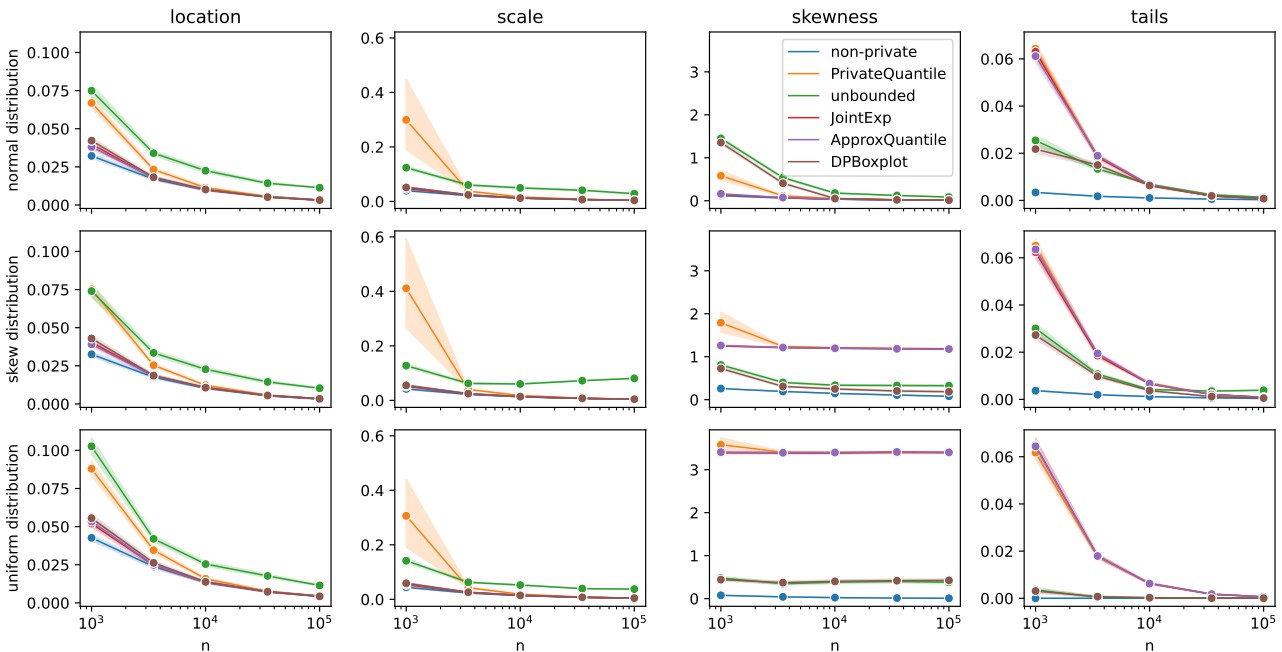

*Figure 5.* Average error in each metric with 95% confidence intervals between the different differentially private boxplots and their population (`oracle`) counterparts (y-axis) with $\epsilon = 0.5$ and increasing $n$ (x-axis). The `DPBoxplot` algorithm is presented, along with the naive method paired with each of the algorithms `JointExp`, `ApproxQuantile`, `unbounded` and `PrivateQuantile` (line color). The different generated distributions are represented row-wise, and the different metrics: location, scale, skewness and tails are presented column-wise. The `non-private` line is the distance between the non-private boxplot and its population counterpart.

## C. Simulation details and additional results

### C.1. Single boxplot estimation

This section presents more details on the simulation study described in Section 5. We simulated data from five distributions, each with mean 0 and variance 1: standard normal distribution (`normal`), skew normal distribution with scale parameter 20 (`skew`), uniform distribution with interval $[-\sqrt{3}, \sqrt{3}]$ (`uniform`), a normalized and mean-centered version of the beta distribution with $\alpha = \beta_n = 2$ (`beta`), and normalized and mean-centered empirical distribution generated from 2019 NY airbnb listing prices. We considered sample sizes of $n \in \{1000, 3500, 10000, 35000, 100000\}$ and privacy budgets of $\epsilon \in \{0.5, 1, 5, 10\}$. Each scenario was simulated 1000 times, leading to simulated data vectors $\mathbf{X}_n$. For each generated dataset, we computed both the non-private boxplot $B(\mathbf{X}_n) = B(\hat{\nu})$ and the population boxplot. We also calculated differentially private boxplots $\tilde{B}(\mathbf{X}_n, \epsilon)$ using each of the methods. Figure 5 and Figure 6 are analogous to Figure 1 in the manuscript, except under $\epsilon = 0.5$ and $\epsilon = 5$, respectively. As anticipated, the performance of all methods decreases with lower $\epsilon$ values, and increases with higher $\epsilon$ values, reflecting the trade-off between privacy and accuracy. However, the variation in performance is marginal in most of the cases, and the same conclusions as those given in Section 5 hold across both privacy levels. In addition, we present Figures 7 and 8, which give a comprehensive summary of the results for `DPBoxplot` from this simulation study.

### C.2. Parameter tuning

We performed simulations under similar settings as those discussed in Section C.1. We varied $\lambda_n \in \{n^{-1/2}, n^{-1/4}, 1\}$, fixing $\epsilon = 1$ and estimated differentially private boxplots with all methods. Figure 9 summarizes the results for simulated distributions (column-wise), boxplot metrics (row-wise). Variations in the method and $\lambda_n$ and are represented with different line colors and line styles, respectively. We observe that none of the methods are highly sensitive to the chosen parameters, except for skewness and tails. Moreover, `DPBoxplot` was generally the best regardless of $\lambda_n$. Based on this observation

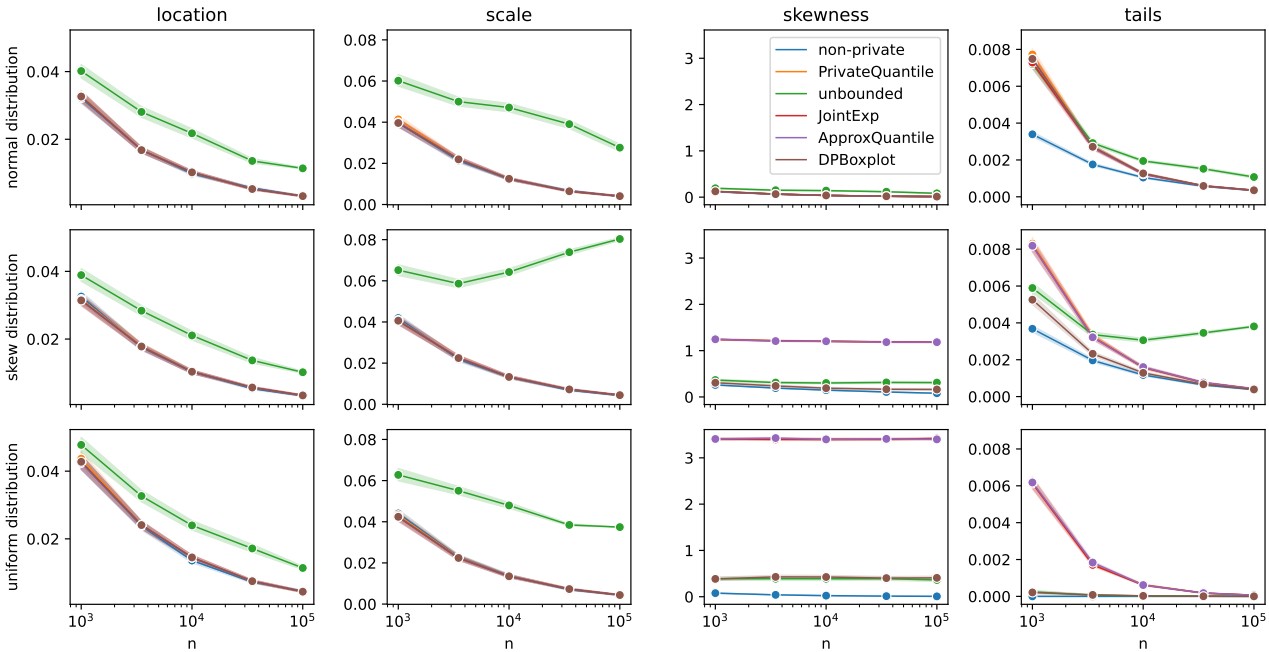

*Figure 6.* Average error in each metric with 95% confidence intervals between the different differentially private boxplots and their population (`oracle`) counterparts (y-axis) with $\epsilon = 5$ and increasing $n$ (x-axis). The `DPBoxplot` algorithm is presented, along with the naive method paired with each of the algorithms `JointExp`, `ApproxQuantile`, `unbounded` and `PrivateQuantile` (line color). The different generated distributions are represented row-wise, and the different metrics: location, scale, skewness and tails are presented column-wise. The `non-private` line is the distance between the non-private boxplot and its population counterpart.

and to account for better results in small samples, we concluded that $\lambda_n = n^{-1/4}$ was a reasonable choice for all methods.

### C.3. Multiple boxplot estimation

This section presents an extensive simulation study to assess the differentially private boxplot's ability to describe, and facilitate comparison of multiple samples. Specifically, we analyze whether the metrics (quantified by distances on location, scale, skewness and tails as in Section 5) between differentially private boxplots across distinct datasets mirror the corresponding pairwise distances observed between their population counterparts. The similitude between two distinct differentially private boxplots is anticipated to align closely with the similitude observed between their respective population counterparts, thereby facilitating analogous visual comparisons and interpretation. For the purpose of quantifying this phenomenon, we introduce the concept of relative similitude between two differentially private boxplots as follows:

$$\tilde{d}\left(\tilde{B}(\mathbf{X}_n, \epsilon), \tilde{B}(\mathbf{Y}_m, \epsilon)\right) = \left| 1 - \frac{d\left(\tilde{B}(\mathbf{X}_n, \epsilon), \tilde{B}(\mathbf{Y}_m, \epsilon)\right) + 1}{d\left(B(\nu_X), B(\nu_Y)\right) + 1} \right|.$$

Here, distance is related to the four metrics previously described. In order to empirically validate the consistency of our proposed approach with this behavior, we conducted Monte Carlo simulations by generating $t$ datasets $(\mathbf{z}_1, \mathbf{z}_2, \cdots, \mathbf{z}_t)$, and two vectors $\mathbf{m}$ and $\mathbf{s}$ of size $t$ sampling from uniform distributions with interval $[-1, 1]$ and $[0.5, 2]$, respectively. Each vector $\mathbf{z}_i$ is size $n_i$ for $i \in \{1, 2, \cdots t\}$ where $n_i$ is chosen randomly such that $n = n_1 + n_2 + \cdots n_t$. Here, $t$ plays the role of the number of treatments in an experiment. We replicated this simulation 1000 times for each combination of $t \in 4, 8$ and $n \in \{500, 5000, 50000\}$ leading to datasets $(\mathbf{X}_1, \mathbf{X}_2, \cdots, \mathbf{X}_t)$ such that $\mathbf{X}_i = s_i \mathbf{z}_i + m_i$. We performed this simulation generating the initial $t$ datasets using mixture of the five distributions described in the previous section such that each distribution generates the same amount of vectors on each replica. For each dataset we calculated non-private boxplots $B(\mathbf{X}_1), B(\mathbf{X}_2), \cdots, B(\mathbf{X}_t)$, and differentially private boxplots using each of the quantile estimation methods and for each

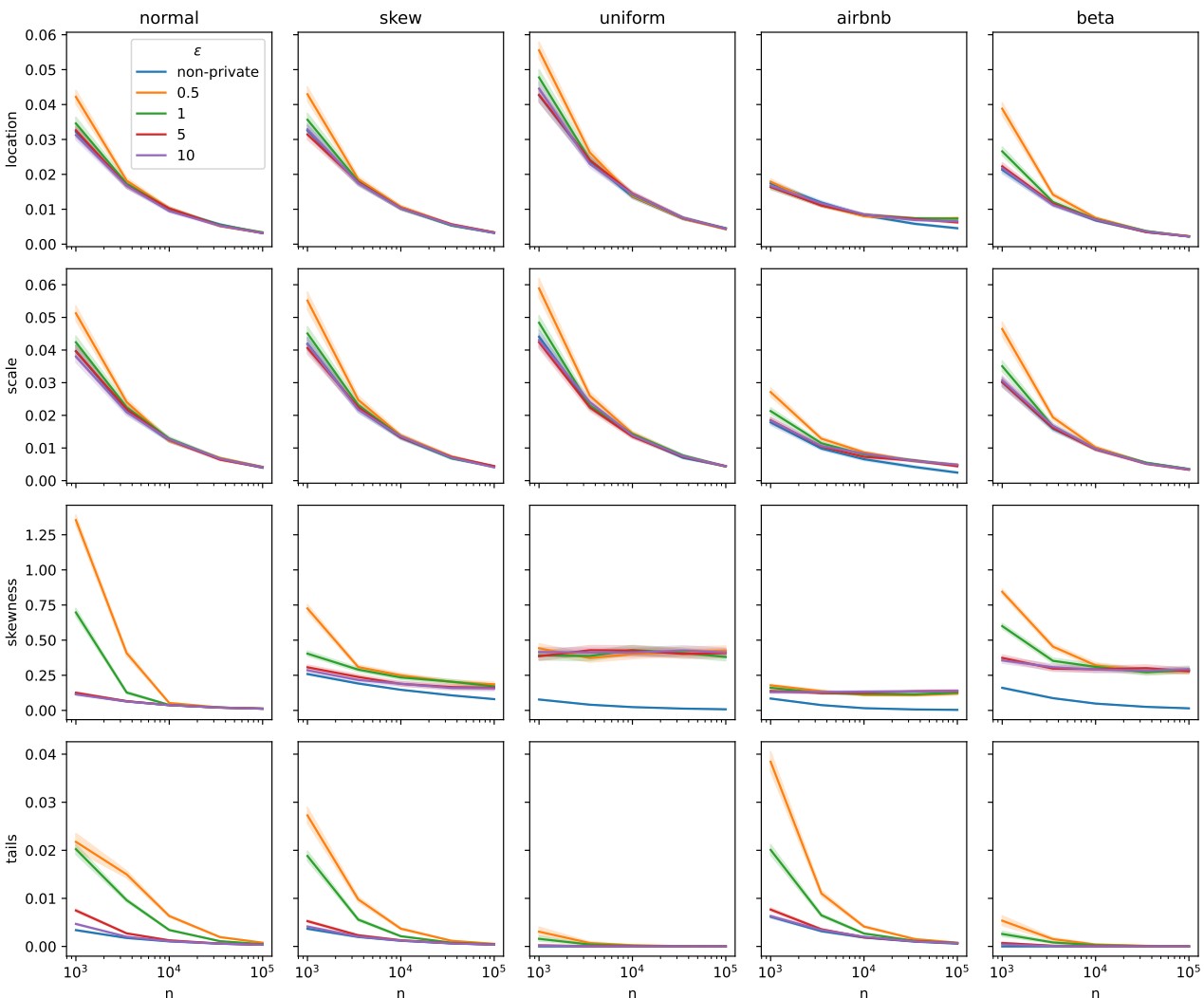

*Figure 7.* Average error in each metric with 95% confidence intervals between estimated differentially private boxplots and `oracle` boxplot (y-axis) with $\epsilon \in \{0.5, 1, 5, 10\}$ (line colors) and increasing $n$ (x-axis); using `DPBoxplot` for private boxplot estimation; from all sampled distributions (column-wise) comparing location, scale, skewness and tails (row-wise). In the legend, `non-private` refers to the distance between the non-private boxplot and the population boxplot.

$\epsilon \in \{0.5, 1, 5, 10\}$. We then calculated pair-wise relative boxplot distances for every pair of differentially private boxplots.

Figure 10 and 11 offers a detailed overview of the results obtained from various scenarios considered. Each boxplot within these plots represents the average pairwise distances observed among all differentially private boxplots generated in a specific scenario. The $x$-axis on each plot denotes the sample size ($n$). Variations in hue color correspond to different methods and different privacy budgets ($\epsilon$), respectively. The first and last two plots in the first row correspond to relative location and relative scale, respectively. The first and last two plots in the second row correspond to relative skewness and relative outliers, respectively. For each pair of relative distance plots, the first and second visualization correspond to $t = 5$ and $t = 10$, respectively. Figure 11 uses $\epsilon = 1$ and Figure 10 shows results with `DPBoxplot` method.

Figure 10 shows that `DPBoxplot` performs consistently well across different scenarios. All other methods have poor performance in at least one scenario. In Figure 11 relative distances exhibit a diminishing trend with augmented sample sizes and $\epsilon$, consistent with expectations illustrated by the `non-private` boxplot. A marginal increase in the parameter $t$ leads to a slight augmentation in the relative distances, attributable to the partitioning of data into smaller subsets and

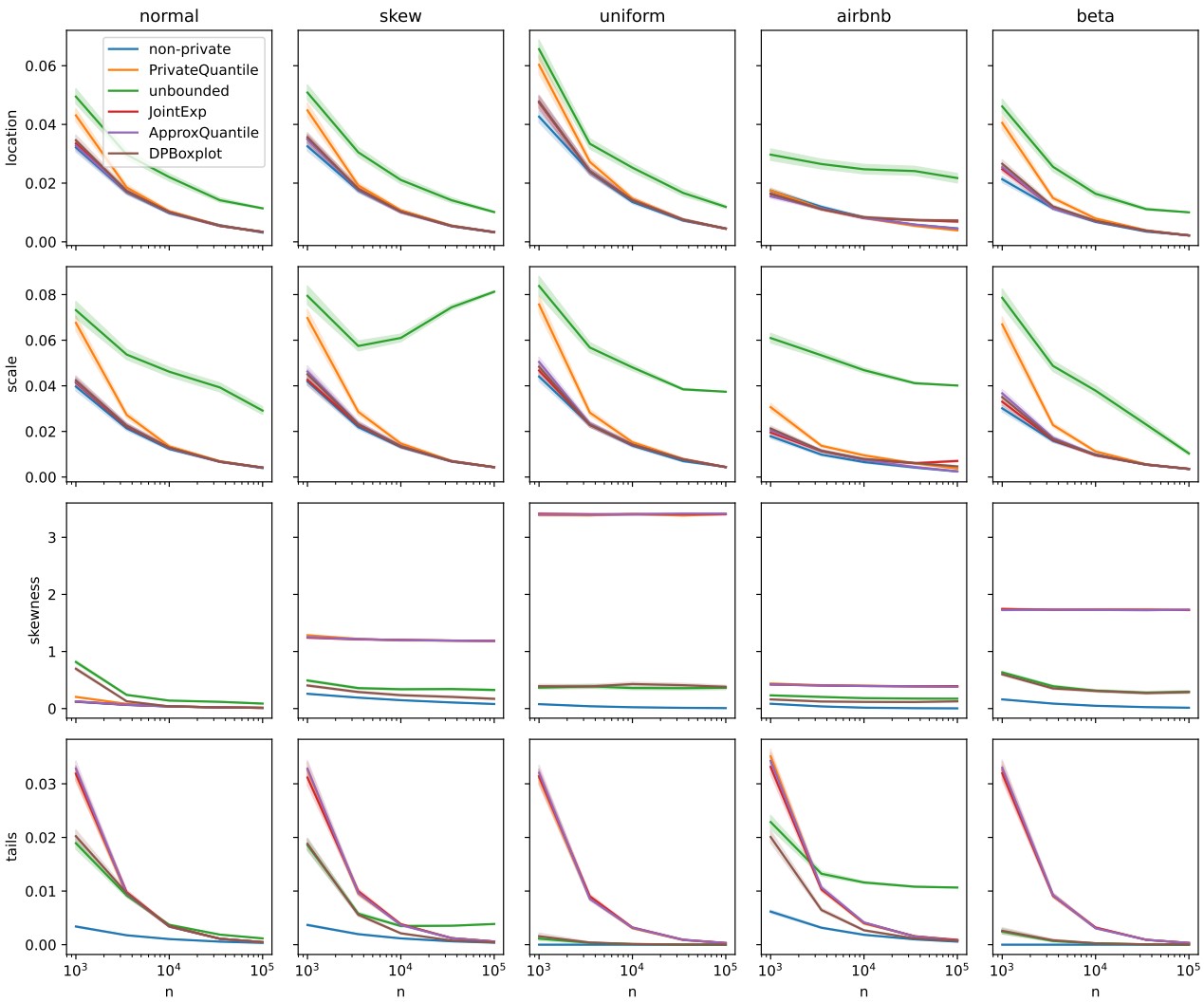

*Figure 8.* Average error in each metric with 95% confidence intervals between estimated differentially private boxplots and `oracle` boxplots (y-axis) with $\epsilon = 1$ and increasing $n$ (x-axis); using boxplot estimation methods `JointExp`, `ApproxQuantile`, `unbounded`, `PrivateQuantile`, and `DPBoxplot` (line colors); from all sampled distributions (column-wise) comparing location, scale, skewness and tails (row-wise). In the legend, `non-private` refers to the distance between the non-private boxplot and the population boxplot.

consequent reduction in sample sizes; however, this impact appears to be negligible. Hence, our analysis suggests that our proposed methodology effectively maintains the visual coherence of relative similarities among diverse boxplots during the execution of multiple comparisons.

### C.4. Computational requirements

All simulations were conducted on a single CPU and did not require significant computational resources. The execution time for the simulations presented in the paper did not exceed 24 hours.

## D. Differentially private quantiles

Here, we include a longer review of the differentially private quantile estimation literature. We focus on the existing statistical results for such estimators, as statistical results for private quantiles is one of our main contributions. It is

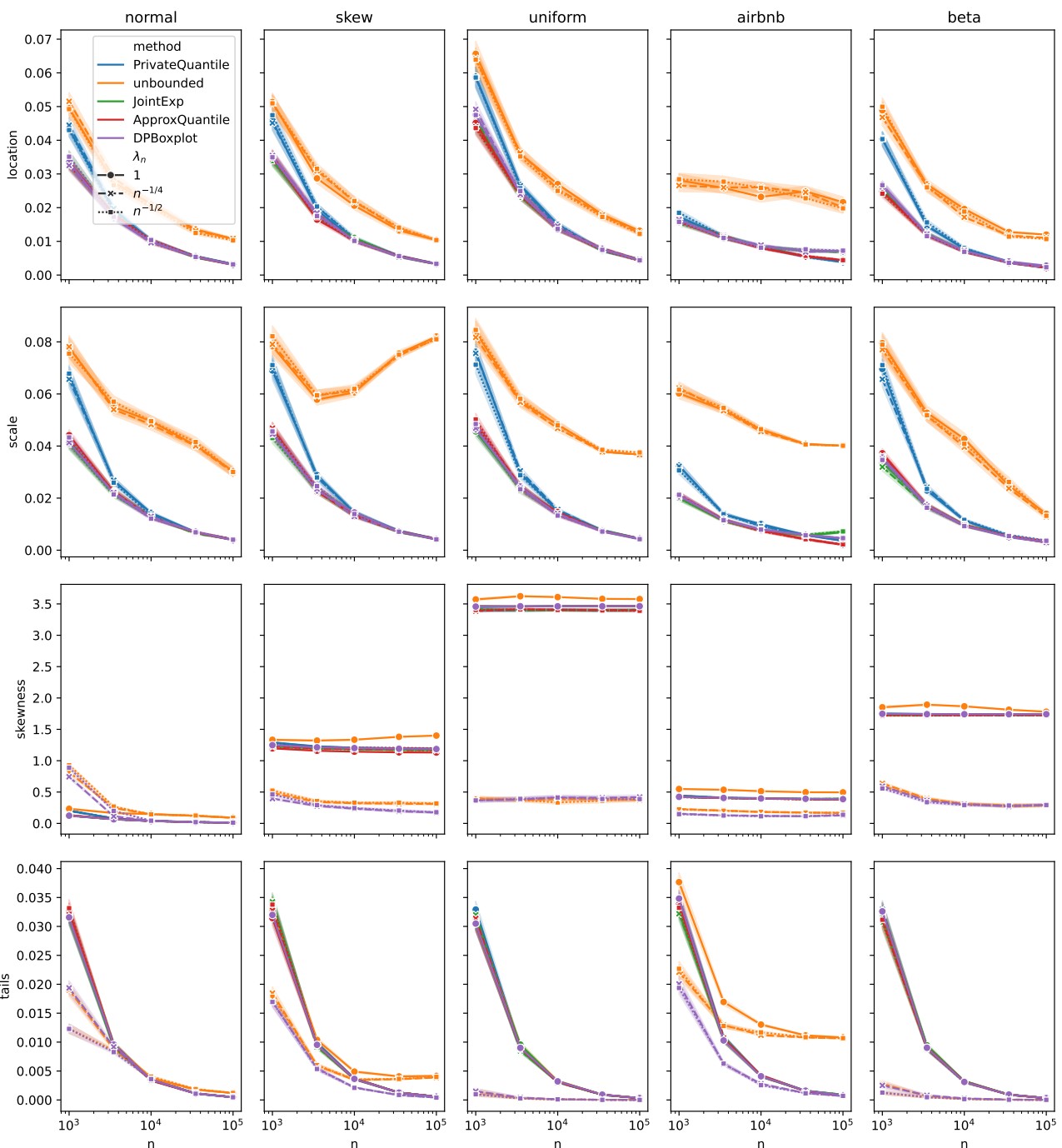

*Figure 9.* Average error in each metric with 95% confidence intervals between estimated differentially private boxplots (y-axis) and `oracle` boxplots with $\epsilon = 1$ and increasing $n$ (x-axis); using various methods (color), with varying $\lambda_n$ (line style); from all sampled distributions (column-wise) comparing location, scale, skewness and tails (row-wise).

also important to mention that range queries can be used to compute quantiles (Bun et al., 2015; Kulkarni, 2019; Kaplan et al., 2020), however, currently, these algorithms may not be very practical, so we do not cover a full review of those. First, Smith (2011) introduced `PrivateQuantile` based on the empirical CDF and the exponential mechanism, and

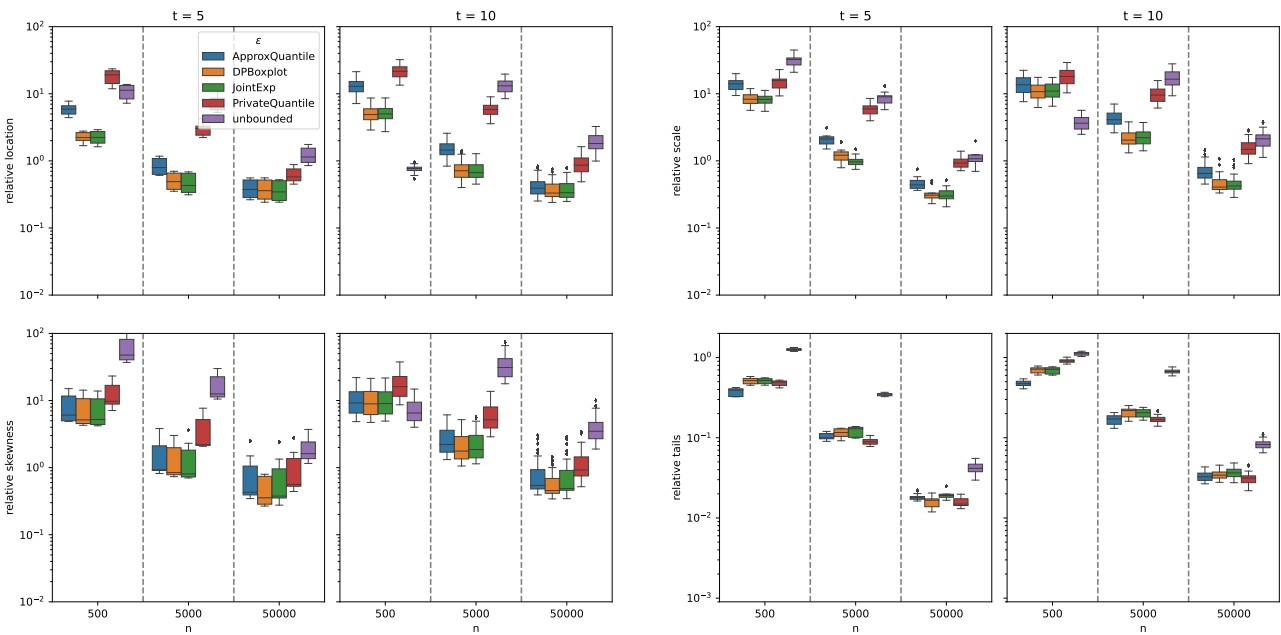

*Figure 10.* Average pair-wise relative boxplot distances (y-axis) between multiple sample boxplots in terms of relative location, scale, skewness and outliers, with $\epsilon = 1$ using different boxplot estimations (color), $n \in \{500, 5000, 50000\}$ (x-axis) and $t \in \{5, 10\}$ (columns). Legend `non-private` is the relative similitude of the non-private boxplots.

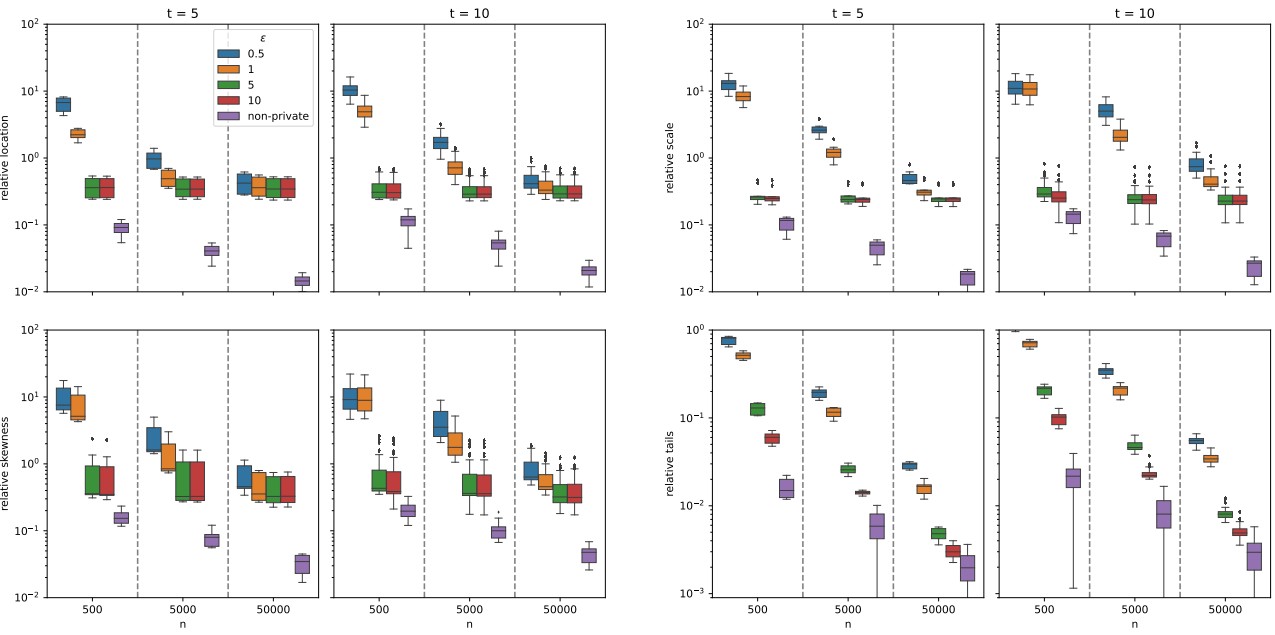

*Figure 11.* Average pair-wise relative boxplot distances (y-axis) between multiple private boxplots in terms of relative location, scale, skewness and outliers, under varying conditions of $\epsilon \in \{0.5, 1, 5, 10\}$ (color), $n \in \{500, 5000, 50000\}$ (x-axis); using `DPBoxplot` quantile estimation and $t \in \{5, 10\}$ (columns). Legend `non-private` is the relative similitude between the non-private boxplots.

present a finite sample accuracy guarantee, under the assumption that the population distribution is close to the normal distribution. An extension of this algorithm is `JointExp`, which can be used to estimate multiple quantiles (Gillenwater

et al., 2021). Later, Lalanne et al. (2023b) and Lalanne et al. (2023a) considered the statistical properties of `JointExp`. Lalanne et al. (2023a) show various concentration results, assuming that the population density has bounded support and is positive in a neighbourhood of the quantiles being estimated. Lalanne et al. (2023b) show that `JointExp` is consistent when the population distribution is continuous, and provide a modified algorithm which can be consistent for population distributions with atoms. This algorithm could also be used to generate the box in the boxplot, if atoms are suspected. Earlier, Asi & Duchi (2020) provided an instance optimal algorithm for the median, based on the inverse sensitivity mechanism. We do not consider this algorithm here because it was shown by Lalanne et al. (2023b) that the algorithm is very similar to `PrivateQuantile`. Similarly, a multiple quantile version based on the inverse sensitivity mechanism is similar to `JointExp` (Lalanne et al., 2023b). Tzamos & Vlatakis-Gkaragkounis (2020) also consider private median estimation, which could be used for private quantile estimation, however, their algorithm takes $n^4$ time, which may be slow in practice, so we did not consider it here. However, we see no reason it would perform poorly to generate the median line in the box. Recently, (Alabi et al., 2022) present a bounded space quantile algorithm, with some concentration bounds that condition on the dataset at hand, and require the sample space to be finite. They also present some statistical guarantees under the assumption of normality. On a related note, some general works imply that no differentially private quantile estimate can have finite sample complexity without making distributional assumptions, .e.g, (Bun et al., 2015). Recently, Durfee (2023) present the `unbounded` algorithm for quantile estimation, which is meant to estimate extreme quantiles well. They do not provide statistical guarantees. Recently, Kaplan et al. (2022) introduce a clever algorithm for estimating multiple quantiles from an existing quantile algorithm, where the statistical error has minimal dependence on the number of quantiles. We call this algorithm paired with `PrivateQuantile` the `ApproxQuantile`. We build on this literature by providing some new statistical results concerning `JointExp`, `PrivateQuantile`, `ApproxQuantile` and `unbounded`. Specifically, we show consistency of the unbounded algorithm under general distributional assumptions, a lower bound on differentiall private quantile estimation for a large class of distributions, and that `JointExp`, `PrivateQuantile`, and `ApproxQuantile` provide minimax optimal, single quantile estimates. We also show that `JointExp`, `PrivateQuantile`, and `ApproxQuantile` are inconsistent for extreme quantiles.

# E. Auxiliary results

**Lemma E.1.** *For all $n \geq 1$, $a, c \in \mathbb{R}$, we have that $n \geq a \log n + c$ if $n \geq 2(a \log a + c) \vee a$.*

*Proof.* First, note that $h(n) = n - a \log n - c$ is increasing for $n \geq a$. Now, for $n \geq a$ we can use the fact that $x - \log x \geq x/2$ for $x \geq 1$ to get $h(n) \geq 0$ whenever $n \geq 2(a \log a + c)$. $\square$

Let $\mathscr{B}$ denote the space of Borel functions from $\mathbb{R}^d$ to $[0, 1]$. For a family of functions, $\mathscr{F} \subseteq \mathscr{B}$, define a pseudometric on $\mathcal{M}_1(\mathbb{R}^d)$, $d_{\mathscr{F}}(\mu, \nu) = \sup_{g \in \mathscr{F}} |\int g d(\mu - \nu)|$, where $\mu, \nu \in \mathcal{M}_1(\mathbb{R}^d)$.

**Definition E.2** ((Ramsay et al., 2024))**.** We say that $\phi(x, \nu)$ is $(K, \mathscr{F})$-*regular* if there exists a class of functions $\mathscr{F} \subset \mathscr{B}$ such that $\phi(x, \cdot)$ is $K$-Lipschitz with respect to the $\mathscr{F}$-pseudometric uniformly in $x$, i.e., for all $\mu, \nu \in \mathcal{M}_1(\mathbb{R}^d)$

$$\sup_x |\phi(x, \mu) - \phi(x, \nu)| \leq K d_{\mathscr{F}}(\mu, \nu).$$

