# OpenReview forum: "Differentially Private Boxplots"
_ICML.cc/2025/Conference — ICML 2025 poster_

### Official Review · Reviewer_VKVL · 2025-03-05

**Overall Recommendation:** 2

**Summary:**

This paper constructs an algorithm for (pure) differentially private box plots by combining two somewhat recent works on private multiple quantiles (for the box), private extreme quantiles (whiskers), and Laplace noise (# outliers). Its theoretical contributions are a few further results about algorithms from these works: a simple lower bound for applying non-extreme private quantiles algorithms to extreme quantiles (Lemma 4.1), a sample complexity upper bound for private multiple quantiles algorithms (Theorem 4.2), a lower bound for a private quantiles (Theorem 4.3), and an asymptotic consistency guarantee for private extreme quantiles (Theorem 4.4) Finally, it shows experimentally that the resulting algorithm, DPBoxPlot, largely outperforms naive approaches based on one quantile algorithm.

## update after rebuttal
After reading the author responses, I'm keeping my score at weak reject. The presented work is a nicely packaged result for a plausible real-world use case, but the algorithmic novelty is IMO too low -- I'm just not that excited by finding a problem that has several pieces and then locating a tool in the literature for each piece. I think the bar for ICML should be higher than that. But I acknowledge that's subjective, so if other reviewers are excited about it, accepting the paper is not bad.

**Claims And Evidence:**

(See other boxes.)

**Essential References Not Discussed:**

None come to mind.

**Experimental Designs Or Analyses:**

I don't see any obvious issues.

**Methods And Evaluation Criteria:**

Sure.

**Other Comments Or Suggestions:**

Some minor stuff:

1)  The notation around the data distribution looks inconsistent. Sometimes it is denoted $\mu$ (start of the exposition of JointExp ) and other times it is $\upsilon$ (end of the exposition of JointExp and most other places). It seems like this should just be $\upsilon$.

2) What is the subscript $1$ in $\mathcal{M}_1(\mathbb{R})$ accomplishing?

3) Line 369 in the experiments says that "[Figure 2's] line style corresponds to the privacy budget $\epsilon$", but $\epsilon$ appears to be fixed to 1 everywhere.

**Other Strengths And Weaknesses:**

Overall, I think the paper's primary strength is that it reasonably answers the question of how to do (pure) DP boxplots. To the best of my knowledge, that answer has not explicitly appeared in the literature. I appreciate that contribution, because boxplots are a basic statistical object, and it's easy to see a practitioner wanting a DP version. The algorithmic answer makes sense, and the experiments are good support.

The paper's primary weakness is that the answer is a straightforward combination of existing work. A boxplot consists of inner and extreme quantiles and outliers (or their counts), and the presented solution combines previous work for each of these. That makes the algorithmic novelty low. The theoretical results help a bit, but one might not really be relevant for DPBoxplot (see #2 in "Theoretical Claims"), Theorem 4.3's main contribution is combining the private Fano's inequality from ASZ21 with the Gaussian approach from TV21, and, subjectively, weak consistency is to me kind of a "valid, but not that helpful" result for extreme quantiles.

This leads me to recommend weak reject. I think the paper contributes something genuine, but I think it's too much a combination of existing work and minor improvements on results about them.

**Questions For Authors:**

(See numbered questions above.)

**Relation To Broader Scientific Literature:**

The paper's key contributions are a bit unusual. Its core algorithm combines existing algorithms in a straightforward way -- if you're aware of those algorithms, this approach is probably one of the first things you'd try. That's a valid contribution, but the primary novelty comes from 1) actually doing the experiments to compare the various plausible methods, and 2) the new results about the existing methods.

**Theoretical Claims:**

1) The text before Theorem 4.2 claims that an upper bound for JointExp implies upper bounds for PrivateQuantile and ApproxQuantile. Why is this? It is clear for the single-quantile case of Lemma 4.1, since the three algorithms become identical, but that isn't true for multiple quantiles. The proof of Theorem 4.2 uses a result, Lemma C.1, that assumes that the vector of estimated quantiles is "a draw from the exponential mechanism" (Appendix Line ~632), but that only holds for JointExp, while the other algorithms are compositions of the exponential mechanism.

2) The text after Theorem 4.2 notes that ApproxQuantile has a utility guarantees with better dependence on $m$. Is that guarantee worse than Theorem 4.2 in another way? If not, why not just use ApproxQuantile instead of JointExp? My understanding from the ApproxQuantile paper is that it's never worse than JointExp, is significantly better for large $m$ (not the setting here, admittedly) and is faster.

---

> ### Author Rebuttal · Authors · 2025-03-28
>
> Thank you for your careful review! We think most of your concern comes from a miscommunication on our part, see point 2 in the following theoretical claims.
>
> **Theoretical Claims:**
>
> 1. The PrivateQuantile algorithm only generates one quantile. So, for PrivateQuantile, we would be applying it $m$ times to estimate $m$ quantiles. Given that all three algorithms are the same with $m=1$, Theorem 4.2 applies to PrivateQuantile with $m=1$. Applying Theorem 4.2 $m$ times with $t=t/m^{1/2}$ results in the same bound (actually a better one in terms of the dependence on m) than is given in Theorem 4.2.  For ApproxQuantile, if you inspect the ApproxQuantile algorithm, it comes from successive applications of PrivateQuantile at each level of the tree, with the input bounds depending on the previous levels of the tree. In that case, you can combine our bound for PrivateQuantile with $m=1$, $t=t/(log m+1)$ and $\gamma=\gamma/m$. Then you can apply Lemma 3.2 of https://arxiv.org/pdf/2110.05429 (Note their notation uses $\beta$ for $\gamma$ and $\gamma$ for $t$) and it follows immediately. Here, you also get a slightly better bound in terms of the dependence on $m$, but here $m = 3$ and so this doesn’t matter too much, especially when we remember that these bounds are stated only up to universal constants which are unknown. We are happy to add a discussion in the appendix about this, so that it is clearer to the reader. We can also add corollaries of Theorem 4.2, if that is helpful.
>
> 2. We have accidentally miscommunicated that a version of Theorem 4.2 exists in the literature for ApproxQuantile with a better dependence on $m$, which is not true. By the sentence “Lastly, we note that the upper bound given in Theorem 4.2 has suboptimal scaling in $m$, the ApproxQuantile algorithm of Kaplan et al. (2022) obtains logarithmic scaling in $m$.” we mean to say that combining our bound with ApproxQuantile results in an improved scaling in $m$. We are just trying to acknowledge that, by combining our bound with their results, we could improve the result in terms of the dependence on $m$, not that there is an identical result in the literature with a better dependence on $m$. We can clarify this in the text. Regarding choosing JoinExp over ApproxQuantile, we observed in simulation that the performance of ApproxQuantile and JointExp was virtually identical, see Figure 2, Figure 5, and Figure 6. This is also easy to see intuitively, where the tree-based solution of ApproxQuantile is not so beneficial if we only end up having a two-level tree. In the end, we went with JointExp because it is easier to understand and implement. The computational complexity is also the same for both algorithms $O(n log(n) + n)$.  We have also made both choices available in the released code, so the user can decide.
>
> **Relation To Broader Scientific Literature:**
>
> We want to highlight that there is little literature on private visualization, as it is tough to quantify the performance of the algorithms in a meaningful way for a data analyst. For instance, no version of a private boxplot had existed, even though DP is not new, and we believe this is because evaluating the performance is challenging. Therefore, one of the main novelties of our work is studying a new direction. For example, the case study is not just a proof of concept, as it sometimes is in other papers. Here, it shows what exactly happens to a boxplot visually in a real data analysis, at different levels of privacy. We are also evaluating if private visualization is even feasible. Therefore, as a first investigation, it makes sense to consider the natural approach in a rigorous way. Furthermore, the most basic approach is the naïve boxplot, which we prove is not a good approach.
>
> **Other Strengths And Weaknesses:**
>
> We agree the algorithmic novelty is low, but the main contribution of our work is providing a statistically valid boxplot that is also verified feasible in a practical setting. Having this tool is essential for many data analysts if they want to do a private data analysis in practice. We respectfully disagree with your comments about the theoretical results.
> -	First, Theorem 4.2 is relevant - we have accidentally miscommunicated that a version of Theorem 4.2 exists in the literature for ApproxQuantile with a better dependence on $m$, which is not true. See the response in “Theoretical Considerations” point 2 above.
> -	We respectfully disagree with this statement: “weak consistency is to me kind of a "valid, but not that helpful" result for extreme quantiles.” This theorem, in combination with the inconsistency of the other algorithms, refutes the naïve boxplot and justifies the proposed boxplot theoretically.
>
> **Minor Comments:** 1 and 3 are typos, thank you. For 2, this is standard notation in probability theory. The 1 represents the space of probability measures and comes from the fact that taking the measure of the whole space gives you back 1.

---

### Official Review · Reviewer_oHSp · 2025-03-13

**Overall Recommendation:** 3

**Summary:**

The paper proposes a new private data visualization in the form of differentially private (DP) boxplots. DP boxplots accomplish this by utilizing three differential privacy mechanisms to privatize the various components of the boxplots – JointExp is used to combute the median and inner quantiles, the unbounded mechanism is used for the whiskers and the Laplace mechanism is used to report outlier counts. The proposed boxplot is benchmarked against the non-private boxplot as the baseline and DP boxplots, where each component is privatized by one of the following DP mechanisms: PrivateQuantile, unbounded, JointExp, ApproxQuantile. The results demonstrate that the proposed private boxplot outperforms all of the other DP boxplot variants on all the statistics (location, scale, skewness and tails).

## Update after rebuttal:
After looking through the other reviews and rebuttals, I concur with the other reviewers that the algorithmic novelty is the primary limitation of this work. Hence, I am staying with my initial score.

**Claims And Evidence:**

The primary claim made is that the proposed DPBoxplot is the best way to create a differentially private boxplot visualization. This claim is furthered in the form of theoretical results that demonstrate that the extreme quantile and outlier estimates are weakly consistent, and the inner quantiles are estimated with optimal sample complexity. These are somewhat backed up by the results on simulation where the DPBoxplot estimates approach the true values of the various statistical measures with more data points ($n$).

**Essential References Not Discussed:**

None that I can think of.

**Experimental Designs Or Analyses:**

Benchmarking is accomplished with the non-private boxplot as the baseline and DP versions of boxplots, wherein the quantiles for each boxplot was privatized by one of several established DP mechanisms. The performance was measured in terms of the average error in the estimates of location, scale, skewness and tails. This analysis on two vales of the privacy budget ($\epsilon$) bear out the claim that DPBoxplot is the most effective way of creating a differentially private box plot. In addition to these analyses, a case study on the Airbnb dataset was included to evaluate the utility of DPBoxplot on real data and its potential limitations.

**Methods And Evaluation Criteria:**

In the absence of a dedicated mechanism to privatize boxplots, the authors have devised reasonable benchmarks, where each of the components of the boxplot is privatized with established DP mechanisms. The non-private boxplot is used as a baseline too.

Evaluation is conducted over the estimated location, scale, skewness and tails of the various boxplots. These are logical criteria as they are the key components of a boxplot; a small error on all these measures is desirable and desirable in any DP boxplot.

**Other Comments Or Suggestions:**

1.	In Section 1, the references in the early part of the second paragraph do not appear to be relevant to the paper; the references should ideally be restricted to data visualization tools.
2.	In the last line on the left side of page 2, ‘sough’ should be emended to ‘seek’.
3.	The variable $q$ could be removed form Theorem 4.2, as $p = q$.
4.	Using a mix of markers and colors in the figures could help distinguish the results for the various algorithms. As it stands, it is difficult to make out some of the lines because they overlap.
5.	Theorems 4.2 and 4.3 appear to be making very similar points. Is Theorem 4.2 simply a step towards arriving at Theorem 4.3, which shows that the inner quantile estimates are consistent.
6.	“Algorithm 3” is mentioned in Section 5 but I am unable to find it in the paper.

**Other Strengths And Weaknesses:**

This paper is largely well-written and contributes novel theoretical results. While there is not much by way of innovation in how the boxplot statistics are estimated privately, the amalgamation of the various DP mechanisms to create a boxplot is in itself a recognizable contribution. I also appreciate the use of Figure 1 as an effective summary of the proposed method and a case study on real data to study the practical utility of the proposed approach.

In addition to this, the description of the DPBoxplot in Section 3 is easy to understand. The discussion of the results in Figure 2 is also comprehensive with the authors acknowledging the lower accuracy of the DPBoxplot estimates at small sample sizes.

However, there are several gaps and unexplained notions used in the paper. Specifically,

1.	What notion of consistency is being used when stating that the inner quantile estimates are consistent? I presume it is the notion of convergence in expectation at large sample sizes but this is not apparent.
2.	What motivated the choice of privacy budget split between the various steps in Algorithm 1?
3.	Why is the parameter $\lambda_n$ required? A clearer justification or demonstration of its utility would be helpful.

**Questions For Authors:**

1.	On line 170, what does $\nu$ refer to?
2.	On line 245, what do $o_{\ell,\nu}$ and $o_{u,\nu}$ refer to?
3.	Theorem 4.3 states a minimax bound on the minimax risk, but the infimum appears to be taken over all DP mechanisms. Doesn’t this mean that the result proves the existence of the a DP mechanism with optimal sample complexity, rather than the optimal sample complexity of the selected mechanism?
4.	In Section 6, the authors state that each visualization is assigned the same privacy budget due to parallel composition. Is this because the selected attribute for each visualization is independent of the rest?

     a.	The paper states that “each visualization is assigned a privacy budget equal to the number of boxplots in the visualization, divided by the number of boxplots on all generated visualizations.” This is difficult to parse. What is the difference between the “visualizations” and “generated visualizations”?
5.	How would DPBoxplots be used in practice? Wouldn’t the repeated visualization of the dataset lead to privacy leakage unless a much stricter privacy budget is used? This may limit its actual utility in practice as typical data exploration entails multiple data visualizations.

**Relation To Broader Scientific Literature:**

Differential privacy is a growing area of research and data visualization is a key component of most practical applications of machine learning. Consequently, if research is to be conducted on private data without formal privacy releases, private data visualizations are an important area of research.

**Theoretical Claims:**

I did not check the proofs of the theoretical claims made in this paper. However, they appear to have been included in the supplementary material.

---

> ### Author Rebuttal · Authors · 2025-03-28
>
> Thank you for your careful review.
>
> **Gaps and unexplained notions:**
>
> 1.	We use weak consistency, otherwise known as convergence in probability. We are happy to clarify this in the paper. This means that if $X_n$ converges in probability to $X$ then for all $t>0$, $Pr(|X_n-X|>t)\to 0$ as $n\to\infty$. This is different from convergence in expectation, otherwise known as $L_1$ convergence.
> 2.	We made this choice based on heuristics, thinking as data analysts. We felt that in practice, it was most important to get the box correct, and so more budget should be allocated to it. The whiskers are more important than the number of outliers, but less than the box. We give the option to change this in our code, depending on the practitioners’ needs, but this would be our recommendation or default parameters.
> 3.	The justification for $\lambda_n$ is as follows: The extreme quantiles are more variable, i.e., estimated less accurately than the inner quantiles. Therefore, without $\lambda_n$, the algorithm is more likely to mistakenly replace the IQR whisker with the extreme whisker. We want to account for this by placing more “trust” on the IQR whisker. Therefore, we add a buffer $\lambda_n$, and instead of replacing the IQR whisker when the extreme quantile is smaller in magnitude, we replace it when it is smaller by at least $\lambda_n$. We can elaborate more in the paper if accepted.
>
> **Other Comments Or Suggestions:**
>
> 1.	We can remove these.
> 2.	Noted, thank you.
> 3.	Here, we are trying to emphasize that Condition 1 holds, with the $p$ in Condition 1 being the $q$ we are referring to in the
> theorem.
> 4.	Thank you, we can add a mix of markers.
> 5.	Theorem 4.2 is the upper bound and Theorem 4.3 is the lower bound. Theorem 4.2 gives an upper bound on the sample complexity of JointExp. Theorem 4.3 gives a lower bound on the sample complexity of any DP estimator for estimating quantiles. The lower bound says that we need at least $n=\Omega…$ samples for any DP estimator to have the risk function less than or equal to $t$.
> 6.	Should be Algorithm 1, sorry about that.
>
> **Questions:**
>
> 1.	$\nu$ is the measure, or distribution, which generates the sample of observations.
> 2.	They are defined there. In words, they are the probability that a random draw from $\nu$ falls below the theoretical lower whisker, and above the theoretical upper whisker, respectively.
> 3.	Theorem 4.3 is concerned with DP mechanisms in general, yes. Theorem 4.3 gives a lower bound on the sample complexity of any DP estimator for estimating quantiles. The lower bound says that we need at least $n=\Omega…$ samples for any DP estimator to have the risk function less than or equal to $t$. Since the upper bound given in Theorem 4.2 for a specific estimator matches the lower bound in Theorem 4.3, we get that the rate is optimal, and no DP estimator can do better.
> 4.	Yes, this is true. To the second question, when we talk about visualizations, we mean one subfigure. For example, in Figure 3, we have three subfigures with 5,3 and 15 boxplots. Therefore, we will assign more budgets to the subfigure that has more boxplots. We can clarify this in the paper if accepted.
> 5.	This is a good question. Yes, we have considered this, but it is a difficult question that we felt was out of the scope of this work. This is the subject of our next investigation.

---

> > ### Comment · Reviewer_oHSp · 2025-04-07
> >
> > Thank you for the clarifications. After looking through the other reviews and rebuttals,I agree that the algorithmic novelty is the primary limitation of this work. Hence, I am satisfied with my initial score.

---

### Official Review · Reviewer_kohb · 2025-03-23

**Overall Recommendation:** 4

**Summary:**

This paper introduces a differentially private algorithm for creating boxplots. The method specializes in the specific quantiles required for a boxplot (median, quartiles, and extremes for whiskers) rather than treating them as a generic sequence of quantiles, as previous differentially private algorithms have done.

The authors propose combining two algorithms: JointExp for estimating inner quantiles (with a new sample complexity analysis) and the “unbounded” quantile estimator from Durfee (2023) for skewness and tails (whiskers), providing consistency results for the extreme values, which they claim is lacking in prior DP quantile algorithms. They also modify the non-private boxplot by reporting the noisy count of outliers instead of plotting them. The authors claim their approach achieves performance similar to non-private methods and outperforms naive private baselines in experiments, including on Airbnb listings data.

**Claims And Evidence:**

- Theorems 4.2 and 4.3 introduce upper and lower bounds for sample complexity of JointExp (proofs are in the appendix) provided that the density is lower bounded in a neighborhoods of the 0.25, 0.5, 0.75 quantiles. This is also supported by Figure 2 that shows estimation improvement of these quantiles with larger sample sizes.

- Theorem 4.4 shows the unbounded algorithm estimator for extreme values is consistent. This is also observed in Figure 2, confirming that their algorithm “retains the best of both worlds”.

- The authors suggest data visualization is underdeveloped in the DP literature; it would be more accurate to state that many DP statistical aggregation techniques can be directly applied to visualization tasks. However, the authors do point out that the specific requirements of common visualizations, like boxplots, have not always been directly addressed with custom algorithms.

**Essential References Not Discussed:**

In addition to [1] above, it might be worth citing recent work on DP median estimation (e.g. [2]), and why not using these algorithms, since they're already aggregating several mechanisms.


[2] Beimel, A., Moran, S., Nissim, K., & Stemmer, U. (2019, June). Private center points and learning of halfspaces. In Conference on Learning Theory (pp. 269-282). PMLR.

**Experimental Designs Or Analyses:**

Results on simulated data report 95% confidence intervals calculated over 1000 trials for each setting. I inspected the code and it contains all the relevant functions for simulations and plotting.

**Methods And Evaluation Criteria:**

The authors use parallel composition to combine two existing DP quantile algorithms into one that works well for inner quantiles and extreme values. They provide mathematical analysis that justify their claims.

They evaluate their method by measuring the error to the real distribution statistics. They compare with previous algorithm for different sample complexities (starting at 1000 samples).  This evaluation is performed on simulated data across different types of distribution providing insights across different scenarios. While their evaluation is thorough enough to guide practitioners, realistic evaluation is not comprehensive (only on the AirBnB dataset) and shows that particular settings, e.g. with small sample sizes, might have inaccurate results.

**Other Comments Or Suggestions:**

- The y-axis labels on the plots show the distribution name instead of the error metric.,
- The paper is missing the code reference in line 403. I assume this is for anonymity reasons.

**Other Strengths And Weaknesses:**

**Strengths**

1. **Novelty**. The paper's focus on the specific quantiles necessary for boxplots, rather than arbitrary quantiles, which is a novel application and contribution for data visualization.

2. **Theoretical Contributions**:
- The paper provides a consistency analysis for extreme quantiles, proving the consistency of the unbounded estimator for whiskers and outliers (Lemma C.3).
- It presents matching (up to logarithmic factors) upper and lower bounds for the sample complexity of JointExp, ApproxQuantile, and PrivateQuantile for inner quantiles under general distributional assumptions (Theorem 4.2 and Theorem 4.3), relaxing previous assumptions of bounded support. The lower bound is novel.

3. **Empirical Evaluation**: The experiments suggest comparable performance to non-private boxplots provided large sample sizes and improvements over naive DP approaches. The application to Airbnb data provides a real-world context.

4. The paper is clearly written, and contributes to the field of differentially private statistics.

**Weaknesses**
1. Clarity of Novelty Regarding DP Data Visualization: The claim that data visualization is underdeveloped in DP is questionable, as it can be seen as a subset of DP statistics.
2. Alternative algorithm selection: The use of the unbounded algorithm for estimating the minimum and maximum (for whiskers) could be better justified. For example,  discussing the literature on private range estimation could make the paper stronger. Similarly, the relationship between JointExp and ApproxQuantile results needs clearer explanation, in particular because the authors claim several times that all the results from JointExp directly apply to ApproxQuantiles.
3. Experimental Detail Omission: It is not clear to me why the authors removed Airbnb listings above $500. This simplifies the problem for all algorithms, yet, only DP-Boxplot performance is shown in the main body.
4. Comparison to Laplace/Gumbel Noise: The authors mention that using Laplace or Gumbel noise made little difference (referencing Durfee (2023)). However, a brief discussion of why exponential noise was chosen, possibly referencing top-K private algorithms results, would strengthen the justification. I think this might be because of the small number of items that this choice does not matter.
5. Complexity Reporting: While the authors mention the time and space complexities are the maximum of the underlying algorithms, explicitly stating these complexities would improve clarity and make the paper self-contained.

**Questions For Authors:**

1.  Regarding the statement, "Given lower and upper bounds on the data a and b... the procedure is still accurate, even when the input bounds are very loose," could you elaborate on the implications of very loose bounds for the performance (e.g., noise level, utility) of the unbounded estimator?
2. Can the authors discuss the selection of budget allocation on Algorithm 1? The authors assign a small privacy budget to outliers, as they "deem these values to be of less interest than the box itself". However, a very noisy count could change conclusions drawn from a plot, particularly with small sample sizes.
3. Throughout the paper, it is stated that results hold for JointExp and consequently for ApproxQuantile. Could you provide a more explicit explanation of why this implication holds?
4. Theorems 4.2 and 4.3 require sufficient distribution mass around q=0.25 and q=0.75. For distributions concentrated around the median, or discrete distributions this might not hold. Do you have suggestions for improving estimation for these quantiles in such (potentially pathological) cases?
5. Regarding Theorem's 4.2 suboptimal dependency on $m$,  authors state it is not relevant since m=3. However, $3/\log(3) \approx 6$ which can be significant in private estimation. Is there a way to match the log(m) bound?
6. The private estimates for "room type by borough" in the Airbnb data show considerable errors. Can you comment on the potential reasons for this and whether any adjustments could mitigate these errors?

**Relation To Broader Scientific Literature:**

This paper builds upon the literature on differentially private quantile estimation (PrivateQuantile (Smith, 2011) , ApproxQuantile (Kaplan et al. 2022)., JointExp (Gillenwater et al. 2021), and unbounded (Durfee 2023)). These works have focused on efficiently estimating multiple quantiles simultaneously; this paper distinguishes itself by focusing specifically on the quantile set required for a standard boxplot (median, quartiles, and extremes for whiskers).

While the unbounded quantile mechanism is used here for estimating the extremes needed for the whiskers, it is worth noting that a separate body of literature exists on differentially private range estimation, such as [1].

[1] Kaplan, H., Ligett, K., Mansour, Y., Naor, M., & Stemmer, U. (2020, July). Privately learning thresholds: Closing the exponential gap. In Conference on learning theory (pp. 2263-2285). PMLR.

**Theoretical Claims:**

Theoretical claims cite previous work and all proofs are provided. The results look intuitive to me but I did not check the proofs in the appendix.

---

> ### Author Rebuttal · Authors · 2025-03-28
>
> Thank you for your careful review!
>
> We are happy to cite the papers on medians and range estimation.
>
> **Weaknesses:**
>
> 1.	DP data visualization poses unique challenges that are not fully addressed by the current literature on DP statistics. For instance, traditional DP statistics focus on numerical accuracy and error bounds, but they do not explore how analysts perceive noisy visualizations. In our paper, in simulation, we consider this by using error metrics based on visuals, and in the case study, where we evaluate what the plots actually look like.
> 2.	This is a good point, we can add a discussion of this literature. In general, unbounded is very practical, easy to implement and runs in linear time, so this was our original motivation . In addition, it is also weakly consistent for the extreme quantiles. We are happy to add an overview of the relationship between ApproxQuantile and JointExp to the appendix. In short, ApproxQuantile applies JointExp with $m=1$ repeatedly. We explain how Theorem 4.2 applies to ApproxQuantile in our response to Reviewer VKVL below.
> 3.	We removed these listings from a data analysis perspective – listings over 500$ a night are idiosyncratic. We only analyze DPBoxplot here to demonstrate its capabilities and see if private exploratory analysis is feasible. The purpose of this section is not to compare the algorithms.
> 4.	In unreported simulations, we tried Laplace, and the performance was comparable.
> 5.	We can add the complexities.
>
> **Questions for the Authors**
>
> 1.	We observe this in simulation, where the bounds are very loose, and the procedure is still accurate. The sample complexity of the unbounded algorithm for extreme quantiles is unknown.
> 2.	We made this choice based on heuristics, thinking as data analysts. We felt that in practice, it was most important to get the box correct, and so more budget should be allocated to it. The whiskers are more important than the number of outliers, but less than the box. We give the option to change this in our code, depending on the practitioners’ needs, but this would be our recommendation or default parameters.
> 3.	See comment 2 in the weaknesses section.
> 4.	This is an interesting question. We have noted that there is a modified algorithm that works for discrete distributions, this was investigated by Lalanne. 2023b. As for the other question, concentrated around the median is fine, but absolutely no density is not fine. When there is no density, this means that we will never sample a point at the quantile, and it cannot be estimated well via empirical quantiles, which are based on sampled points. In that case, we would have to use something other than the empirical distribution or CDF to estimate the quantile, such as a parametric model.
> 5.	See the comment below to Reviewer VKVL below in point 1, where if we apply our methods to ApproxQuantile we get poly-logarithmic dependence on $m$. Observe though that these bounds are all stated up to constants, so, while $\log m +1$ is smaller than $m$, the difference in constants between ApproxQuantile and JointExp may counteract this.
> 6.	We answer this at the end of Section 6. See the text “These discrepancies...” In terms of remedies, given that the sample size was only 9, there is not much that can be done.

---

### Decision · Program_Chairs · 2025-05-01

**Decision:**

Accept (poster)

**Comment:**

Although the algorithmic novelty of this paper is somewhat low, it nevertheless offers a thorough, clear, and and convincing argument for the authors' proposed method for generating differentially private boxplots. The method appears to behave nicely across a range of experiments, and the theoretical results, though not revolutionary, underline the algorithmic choices. The authors provide code, and any practitioners in need of private boxplots would likely benefit from the work.

The authors are encouraged to use the additional page allowed in the final version to address reviewer comments; for example, they can clarify the relationship to the literature on range estimation, more clearly explain the role of $\lambda_n$, and perhaps give more intuition around Theorems 4.2 and 4.3, which were confusing to several reviewers.